METHODS AND RESOURCES

# A noninvasive BCG skin challenge model for assessing tuberculosis vaccine efficacy

**Nitya Krishnan**[1], **Miles Priestman**[1], **Iria Uhía**[1], **Natalie Charitakis**[1], **Izabella T. Glegola-Madejska**[2], **Thomas M. Baer**[3], **Albin Tranberg**[4], **Alan Faraj**[4], **Ulrika SH Simonsson**[4], **Brian D. Robertson**[1]*

1 Department of Infectious Disease, Imperial College London, London, United Kingdom, 2 Department of Life Sciences, Centre for Bacterial Resistance Biology, Imperial College London, London, United Kingdom, 3 Stanford Photonics Research Center, Stanford University, Stanford, California, United States of America, 4 Department of Pharmaceutical Biosciences, Uppsala University, Uppsala, Sweden

* b.robertson@imperial.ac.uk

**Data Availability Statement:** Relevant data are within the paper and its Supporting Information files. Flow cytometry FCS files are available from

## Abstract

We report here on the characterisation in mice of a noninvasive bacille Calmette-Guérin (BCG) skin challenge model for assessing tuberculosis (TB) vaccine efficacy. Controlled human infection models (CHIMs) are valuable tools for assessing the relevant biological activity of vaccine candidates, with the potential to accelerate TB vaccine development into the clinic. TB infection poses significant constraints on the design of a CHIM using the causative agent *Mycobacterium tuberculosis* (Mtb). A safer alternative is a challenge model using the attenuated vaccine agent *Mycobacterium bovis* BCG as a surrogate for Mtb, and intradermal (skin) challenge as an alternative to pulmonary infection. We have developed a unique noninvasive imaging system based on fluorescent reporters (FluorBCG) to quantitatively measure bacterial load over time, thereby determining a relevant biological vaccine effect. We assessed the utility of this model to measure the effectiveness of 2 TB vaccines: the currently licenced BCG and a novel subunit vaccine candidate. To assess the efficacy of the skin challenge model, a nonlinear mixed-effects models was built describing the decline of fluorescence over time. The model-based analysis identified that BCG vaccination reduced the fluorescence readout of both fluorophores compared to unvaccinated mice ($p <$ 0.001). However, vaccination with the novel subunit candidate did not alter the fluorescence decline compared to unvaccinated mice ($p >$ 0.05). BCG-vaccinated mice that showed the reduced fluorescent readout also had a reduced bacterial burden in the lungs when challenged with Mtb. This supports the fluorescence activity in the skin as a reflection of vaccine induced functional pulmonary immune responses. This novel noninvasive approach allows for repeated measurements from the challenge site, providing a dynamic readout of vaccine induced responses over time. This BCG skin challenge model represents an important contribution to the ongoing development of controlled challenge models for TB.

https://doi.org/10.5281/zenodo.12794251. Python scripts are available from https://doi.org/10.5281/zenodo.12781429.

**Funding:** BDR received funding from Aeras grant "Human Challenge Model for TB". BDR and TMB received funding from the Bill and Melinda Gates Foundation grant OPP1180610. The sponsors played no role in study design, data collection and analysis, decision to publish, or preparation of the manuscript.

**Competing interests:** The authors have declared that no competing interests exist.

**Abbreviations:** BCG, bacille Calmette-Guérin; CFU, colony-forming unit; CHIM, controlled human infection model; FOCEI, first-order conditional estimation method with interaction; IAV, inter-animal variability; ID, intradermally; Mtb, *Mycobacterium tuberculosis*; OFV, objective function value; PsN, Perl-speaks-NONMEM; RFU, relative fluorescence unit; SCM, stepwise-covariate modelling; TB, tuberculosis; VPC, visual predictive check; YFP, yellow fluorescent protein.

## Introduction

Despite advances in modern medicine and better living conditions for many people, tuberculosis (TB) remains a major killer of the poor and disadvantaged around the world. Notwithstanding major efforts in the diagnosis, treatment, and prevention of TB—which have saved an estimated 54 million lives between 2000 and 2017—these have had little impact on transmission, and there are still nearly 10 million newly diagnosed cases and around 1.5 million deaths every year [1]. The current bacille Calmette-Guérin (BCG) vaccine is routinely given to neonates in endemic countries and high-risk populations and protects against the disseminated forms of TB disease to which children are susceptible. However, BCG works poorly in adolescents and adults who are the significant drivers of TB in high incidence communities [2,3], meaning that large-scale vaccination programmes have had negligible impact on transmission. Consequently, we need new vaccines with efficacy in all ages and all populations to successfully control TB and reach the WHO End-TB targets of a 95% decrease in deaths and a 90% decrease in incidence by 2035 (both relative to 2015) [4].

We have learnt much about the pathogenesis and immunology of TB using the mouse model. However, TB vaccine discovery has been hampered by animal models that poorly reflect the stages of infection and disease found in humans. Candidate vaccines that show good efficacy in a range of animal models do not always perform as well in human trials, where a lack of biomarkers for vaccine efficacy make incident TB the primary outcome, which requires several years of follow-up post-vaccination [5,6]. As the number of vaccine candidates completing preclinical studies increases [7–10], there is a need to prioritise those that proceed to expensive large-scale clinical trials without relying solely on the current suite of animal model data.

Human infection/challenge models have a long history in experimental medicine [11]. The malaria sporozoite challenge model has a safe history of use to assess new drugs and vaccines with a relatively short antimalarial treatment at the conclusion of the study [12,13]. A human challenge model for TB creates several challenges, including pulmonary infection, lengthy 6-month drug treatment, and the potential for latent or asymptomatic infection. The establishment of a pulmonary human infection model using BCG has demonstrated its feasibility for advancing the understanding of pulmonary immunobiology during TB infection [14]. A strategy to avoid issues associated with *Mycobacterium tuberculosis* (Mtb) as a challenge agent is to use *Mycobacterium bovis* BCG, which has a long history of safe use and has been used as a surrogate [14–18]. Our study utilises BCG as the basis for a fluorescent reporter strain (FluorBCG) that can be introduced intradermally, with the fluorescent signal measured noninvasively through the skin using a sensitive and cost-effective imaging system. This system reproducibly measures an accelerated loss of fluorescent signal over time in vaccinated compared to nonvaccinated animals. In this report, we describe the development and characterisation of the system and demonstrate its ability to detect relevant vaccine effects in the mouse model of TB. This skin-based challenge model has the potential to be used as an early indicator of vaccine efficacy and provide useful data to inform and advance vaccine development. This feasibility study is a step towards developing a human challenge model for TB.

## Materials and methods

### Animal studies

All animal procedures were performed under the Animal Scientific Procedures Act (1986) under the licence issued by the UK Home Office (PPL 70/7160). Six- to 8-week-old female BALB/c mice (Charles River, United Kingdom) were maintained in Biosafety Containment Level 3 facilities (BSL3) according to institutional protocols.

## Bacterial strains and growth conditions

Experiments with Mtb H37Rv (a kind gift from Dr Christophe Guilhot, Institut de Pharmacologie et de Biologie Structurale, France) were carried out in BSL3 facilities according to institutional protocols. BCG-Pasteur and BCG Pasteur ΔpanCD strains were a kind gift from Dr Nathalie Cadieux, Aeras Global TB Vaccine Foundation. Mycobacterial strains were cultured in Difco Middlebrook 7H9 liquid medium (Becton Dickinson, UK) supplemented with 0.2% glycerol (Sigma, UK), 0.05% Tween 80 (Sigma, UK), and 10% oleic acid-albumin-dextrose-catalase (OADC, USBiological, UK). For growth on solid medium, Middlebrook 7H10 plates were supplemented with 0.5% glycerol and 10% OADC. BCG Pasteur ΔpanCD was grown in medium supplemented with 24 µg/ml calcium pantothenate (Sigma-Aldrich, UK).

## Construction of the fluorescent mycobacterial strain

We constructed a recombinant fluorescent mycobacterium, BCG Pasteur ΔpanCD [Psmyc Turbo635asv-YFP] (FluorBCG), expressing dual fluorophores (one unstable to increase detection of growth/death). The pCB22 backbone plasmid (6572 bp; Dr Nathalie Cadieux, Aeras Global TB Vaccine Foundation) is a shuttle vector containing *E. coli* and mycobacterial origins of replication, a hygromycin resistance cassette, and the *panCD* operon driven by the constitutive BCG_3667c promoter; the plasmid complements the *panCD* auxotrophy in the host BCG ΔpanCD strain ensuring the plasmid is stably maintained without antibiotic selection. Turbo635 has been previously used for in vivo imaging of mycobacteria [19], and Turbo635asv has a C-terminal degradation tag added to promote protein turnover [20]. The monomeric superfolder (msf) yellow fluorescent protein (YFP) is a derivative of GFP [21]. An artificial operon was constructed with both genes driven by the Psmyc mycobacterial strong promoter [22] and cloned into the XbaI-PacI sites in the pCB22 MCS downstream of the *panCD* operon (S1 Fig). The fluorescent BCG reporter strain was obtained by electroporation of the plasmid into BCG Pasteur ΔpanCD as previously described [23].

A dual fluorescent reporter strain of Mtb was also constructed by transforming the same Psmyc Turbo635asv-YFP reporter plasmid into an H37Rv *panCD* mutant kindly provided by Prof Bill Jacobs, Albert Einstein College of Medicine to create Fluor-Mtb.

## Immunisation and challenge

Mice were vaccinated subcutaneously with $1 \times 10^4$ BCG Pasteur and rested for 4 weeks before challenge. In Mtb challenge groups, each mouse was infected with $1 \times 10^3$ colony-forming units (CFUs) of strain H37Rv in 35 µl via the intranasal route. For the fluorescent strain challenge, either $5 \times 10^6$ CFU of FluorBCG or $5 \times 10^6$ CFU of Fluor-Mtb was injected intradermally (ID) with a 30G insulin needle (Easy Touch, United States of America) into the skin on the dorsal side of the mouse ear. Ears were imaged immediately post-challenge to provide a reading for baseline fluorescence; this was the day 0 time point. Imaging of the ears was repeated at regular intervals until day 28. Four weeks post-challenge with Mtb, the bacterial burden was determined in the lungs and spleen. Lungs and spleen were removed aseptically, homogenised in PBS containing 0.05% Tween 80, and serial dilutions of the organ homogenates were plated on Middlebrook 7H10 agar plates. The number of CFU was enumerated 21 days later. A summary of experiments and treatments is provided in S1 Table.

## ChAdOx1.PPE15 strain and immunisations

ChAdOx1.PPE15 was a kind gift from Dr Elena Stylianou and Prof Helen McShane, University of Oxford [24]. Female BALB/c mice were immunised intranasally with $1 \times 10^8$ infectious

units of the virus in a final volume of 35 μl. A summary of experiments and treatments is provided in S1 Table.

## Portable imager

The portable imager instrument was designed to provide quantitative images of the fluorophore emission from intradermal injections of FluorBCG in the mouse ears. FluorBCG is excited by green (Cree XPE2, Green) and amber (Cree XPE2, Amber) LEDs, which are chosen to have emission maxima near the peak excitation of the YFP and Turbo635 fluorophores, respectively. The LEDs broad band emissions are narrowed using dichroic filters centred at 505 nm (YFP) and 590 nm (Turbo-365) with passbands of 20 nm. The Nikon D5300 camera exposure settings are typically ISO 400 and shutter speed 1/30 second. The images are captured in RAW format and then are converted to 48-bit TIFF images for quantitation. The images are analysed using custom software that integrates the fluorescent intensities over a user-specified region, typically at 4 mm circular region centred on the intradermal injection location.

## Construction details of the portable imager

The imager consists of a Nikon 5300 SLR colour camera with a 24.2-megapixel sensor with a 3.89-μm pixel pitch, and a 14-bit resolution depth. The image is formed on the camera sensor using an image relay system consisting of a 75-mm focal length, 25-mm diameter objective lens and a 100-mm focal length, 50-mm diameter tube lens. The field of view of the imaging system is approximately 18 mm by 12 mm. The mouse ear is located at the focus of the objective lens and the camera sensor is located at the focal plane of the tube lens. The mouse is located in a sealed chamber with a transparent cover. The mouse ears are gently held flat in the chamber and are positioned to be at the image plane of the camera. The filtered LED broad band emission of YFP (505 nm) and Turbo635 (590 nm) is partially collimated by a 25-mm focal length, 25-mm diameter lens located approximately 25 mm from the LED source. Typical LED intensities at the image plane are 20 mW/cm$^2$ at 505 nm and 25 mW/cm$^2$ at 590 nm. The fluorescent emission from the bacteria passes through a dichroic filter placed between the objective lens and the tube lens with passbands centred at 542 nm and 639 nm, with passband widths of 27 nm and 42 nm, respectively.

## Python pipeline

A semiautomated, user-friendly python pipeline was created for data analysis. The pipeline requires python 3.7.0 to be installed and packages: numpy 1.20.1, matplotlib 3.0.2, pandas 0.23.4, scipy 1.6.2, and xlsxwriter 1.3.8. Inputs for the pipeline are the text files generated from the custom-built imager containing the average intensity for YFP and Turbo635 fluorescence. The imager software has established analytical routines for measuring the average fluorescence over a defined area and storing this information as text files rather than images, allowing for a less memory intensive pipeline input. It is possible to collect background measurements of fluorescence and subtract these values from the measurements of the injected area to decrease background noise. While different visualisations can be generated using the pipeline, the initial processing steps are consistent. First, as images of each ear are taken separately, these fluorescence intensity values are averaged for each mouse and the standard deviation for all mice at each time point is calculated. The pipeline can visualise any number of mice and time points but assumes there is a maximum of 2 fluorescence colours. In addition to plotting the average fluorescence intensity value for all mice at each time point, the pipeline can also plot the normalised average values, where each time point after the first recorded fluorescence value is

calculated as a percentage of the initial value. The code generated for the python pipeline is available here: https://doi.org/10.5281/zenodo.12781429.

## Isolation of cells from murine ear

Mouse ears were removed postmortem using dissecting scissors and cut into small pieces to facilitate tissue digestion. The tissue pieces were incubated in a digestion cocktail of 300 μg/ml Liberase TM (Roche, UK) and 50 U/ml of DNaseI (Sigma-Aldrich, UK) at 37°C with slow agitation for 90 min [25]. The digested skin fragments were passed through a nylon 100 μm cell strainer (Falcon, Thermo Fisher Scientific, UK) to obtain single-cell suspensions. Cell viability was >90% as determined by Trypan Blue (Sigma-Aldrich, UK) exclusion.

## Flow cytometry and cell sorting

Red blood cells were lysed using RBC lysing buffer (Sigma-Aldrich, UK). Cells were subsequently washed, enumerated, and the cell concentration adjusted to $1 \times 10^7$ cells/ml in PBS (Thermo Fisher Scientific, UK). The single-cell suspension was incubated with TruStain FcX anti-mouse CD16/CD32 antibody (BioLegend, UK) for 10 min at 4°C. Following Fc block, cells were stained using horizon fixable viability stain 510 (Becton Dickinson, UK) for 15 min at room temperature, washed, and then resuspended in staining buffer (PBS +1 mM EDTA (Invitrogen, UK) +0.1% BSA (SigmaAldrich, UK)). Cells were then surface stained with fluorochrome-conjugated antibodies summarised in Table 1. Following antibody staining, cells were washed, resuspended in PBS, filtered through a Falcon tube with an integrated cell strainer (Fisher Scientific, UK) before being analysed and sorted using the FACSAria III (Becton Dickinson, UK) cell sorter housed in a biosafety cabinet. Flow cytometry data were acquired using the FACSDiva software. The gating strategy used to identify the immune subsets in the ear are presented as supporting data (S5 Fig). Post-acquisition analysis was performed using FlowJo 10.7.1 (Becton Dickinson).

## Statistical analysis

To assess the efficacy of the skin challenge, 2 nonlinear mixed effect models were built to, empirically, describe the decline of fluorescence over time. The model was based on data collected from 5 experiments with a total of 100 mice (S1 Table), including 45 vaccinated with BCG, 45 unvaccinated, and 10 vaccinated with ChAdOx1.PPE15. Eighty mice received the FluorBCG reporter, and 20 Fluor-Mtb. There were 1,030 observations in total. Among these, 455 observations were from unvaccinated mice, 455 observations were from BCG-vaccinated

**Table 1. List of antibodies used for flow cytometry.** All antibodies were purchased from BioLegend, UK.

| Antibody | Fluorophore | Clone |
|---|---|---|
| Anti-CD45 | APC/Cy7 | 104 |
| Anti-Ly6G | PE/Cy7 | 1A8 |
| Anti-CD11c | BV785 | N418 |
| Anti-CD11b | BV650 | M1/70 |
| Anti-I-A/I-E | AF700 | M5/114.15.2 |
| Anti-CD64 | BV711 | X54-5/7.1 |
| Anti-F4/80 | BV605 | BM8 |
| Anti-LyC | BV421 | HK1.4 |
| Anti-CD103 | BV421 | 2E7 |
| Anti-CD207 | PE | 4C7 |

mice, and 120 observations were from ChAdOx1.PPE15-vaccinated mice. Forty-five mice underwent pulmonary challenge with Mtb. This subset included 20 mice vaccinated with BCG, 20 unvaccinated mice, and 5 mice vaccinated with ChAdOx1.PPE15. The median and range of log10_fluoroscence from the YFP reporter was 8.754 (min: 6.326, max: 9.476). Similarly, for the Turbo635 reporter, the median and range was 7.575 (min: 6.847, max: 8.836). Twenty animals received the Fluor-Mtb reporter, of which 10 were vaccinated with BCG, and 10 unvaccinated. A total of 220 observations were collected from the Fluor-Mtb reporter animals.

Separate models were developed for each fluorophore using log transformed data. Different structural models were evaluated (Eqs 1 and 2) to find a base structural model to empirically describe the 2 endpoints.

$$log_{10}RFU = log_{10}(e^{intercept1 - SLOPE*TIME}) \tag{1}$$

$$log_{10}RFU = log_{10}((e^{intercept1 - SLOPE1*TIME}) + (e^{intercept2 - SLOPE2*TIME})) \tag{2}$$

An additive residual error model was used. Thereafter, all combinations of inter-animal variability (IAV) were explored on the different parameters of the base model. The individual parameters were assumed to be log-normally distributed and covariance between different IAV were explored. Furthermore, Box-cox transformation of the IAV was evaluated on all parameters.

The final base model was taken forward to study the effects of available covariates (vaccination status, pulmonary challenge status, reporter, and baseline fluorescence) using stepwise-covariate modelling (SCM) [26]. The SCM was configured to include a covariate on a parameter if it would result in an objective function value (OFV) drop corresponding to a statistical significance change of $p \leq 0.05$ during the forward step. For the backward step, it was configured to only remove covariates if the removal corresponded to a statistically significant increase of the OFV using a stricter criterion of $p \leq 0.01$. Only the reporter covariate was allowed to be implemented on the first intercept parameter, while all covariates were allowed to be implemented on the remaining parameters.

Parameter estimation was performed using nonlinear mixed effects modelling in NON-MEM (version 7.5.1; ICON plc, North America, Gaithersburg) [27]; the first-order conditional estimation method with interaction (FOCEI) was used [28]. Perl-speaks-NONMEM (PsN) (version 5.3.1, Department of Pharmacy, Uppsala University, Sweden) was utilised to run models, generate visual predictive checks (VPCs), and SCM runs [29].

Model evaluation was performed by comparing the OFV of nested hierarchical models, where a decrease in OFV of 3.84 can be considered statistically significant at a 5% level for one degree of freedom change using a $\chi^2$ distribution. In addition, stratified VPCs, goodness of fit plots, precision of model parameters, shrinkage, scientific plausibility, and model stability were considered in the model selection and evaluation procedure. Graphical and numerical analysis of the data was performed in R (version 4.2.3, R Foundation for Statistical Computing, Vienna, Austria). All model evaluation plots were generated in R package Xpose4 (version 4.7.2; Department of Pharmacy, Uppsala University, Sweden). Documentation and comparison between models were performed using Pirana (Version 23.1.1, build 1, Certara, Princeton, USA).

The final model for each fluorophore was used to simulate the typical predictions (no IAV or residual error) over 30 days to visualise the covariate effects found on the measured endpoints fluorescence over time.

## Results

### Fluorophore selection

There are 2 issues to consider when imaging fluorescent reporters in vertebrates. Firstly, the ability of light to penetrate tissue is a function of its wavelength, with red light above 600 nm showing less light absorption by haemoglobin [30]. Secondly, fluorescent proteins are stable, so signals take time to decay after the cell stops producing the fluorophore. To address these issues, we decided to construct a dual reporter strain, FluorBCG, including an unstable version of a red fluorophore that would ensure tissue penetration and be degraded faster as the bacteria stop growing and die, plus a stable fluorophore protein as an additional marker of bacteria should degradation of the unstable fluorophore take it below detectable levels. We surveyed the available fluorophores for those reported to be bright and stable in mycobacteria. We have previously demonstrated the utility of the red fluorophore Turbo635 for imaging in murine systems [19] and made an unstable derivative by the addition of the tripeptide ASV to the C-terminus [20]. We selected a YFP reported as bright and stable in mycobacteria [21]. A YFP is a monomeric superfolder derivative of GFP that folds faster and more efficiently with superior solubility and brightness [31]. Fig 1A shows each fluorophore was detected by the custom imager, but Turbo635ASV signals were noticeably dimmer. Quantification showed 10-fold less Turbo635ASV RFUs compared to YFP (S2 Fig).

### Utilising fluorescence output as a measure of vaccine efficacy in a murine skin challenge model

We chose the skin of the mouse ear as a suitable model to establish proof-of-concept for noninvasive monitoring of vaccine responses using a fluorescence-based readout. The mouse ear is thin with little fur or pigmentation, making it ideal for imaging and avoiding the autofluorescence associated with fur. A dose escalation study was carried out to determine the bacterial dose that would provide the optimum signal to noise ratio for imaging. Female BALB/c mice were dosed with 3 different concentrations of fluorescent BCG (FluorBCG): low dose of $5 \times 10^4$ CFU, mid-dose of $5 \times 10^5$ CFU, and high dose of $5 \times 10^6$ CFU. The high dose inoculum provided the optimum signal to noise ratio for YFP and was well tolerated in mice with no adverse effects (S2 Fig). Female BALB/c mice were then vaccinated with BCG and 4 weeks later challenged ID in each ear with $5 \times 10^6$ CFU of FluorBCG. Control mice were unvaccinated. Fluorescence readings were taken at the indicated time points until 28 days post-challenge (Fig 1A). A time-course experiment captures the YFP fluorescence dynamics observed in the control and BCG-vaccinated groups (Fig 1B and 1D). Data were normalised to day 0 to allow for operator-dependent variations in the initial intradermal inoculation in the ear (Fig 1C and 1E). The initial YFP readout (days 0 to 2) was similar in both vaccinated and control groups. From day 6 to day 21, there is a marked difference in groups, with a lower YFP readout in the vaccinated compared to control mice. After day 21, the YFP output declines in both vaccinated and control groups (Fig 1B–1E). In the control group, the increase in YFP fluorescence on day 5 and day 6 could represent a short growth period. In contrast, in the BCG-vaccinated group, the fluorescence readout declines throughout the measurement period. There are some variations in signal, but these are mirrored in both vaccinated and control groups suggesting a measurement anomaly on those days. The YFP fluorescence kinetics in the vaccinated and control groups are paralleled in the Turbo635 output, with the only major difference being a 10-fold decrease in relative fluorescence units (RFUs) (S3 Fig).

### Fluorescence output related to viable BCG burden in the ears

Fluorescent proteins have relatively long half-lives, which could result in false-positive signals from residual fluorescent proteins when bacteria are no longer viable. To investigate BCG

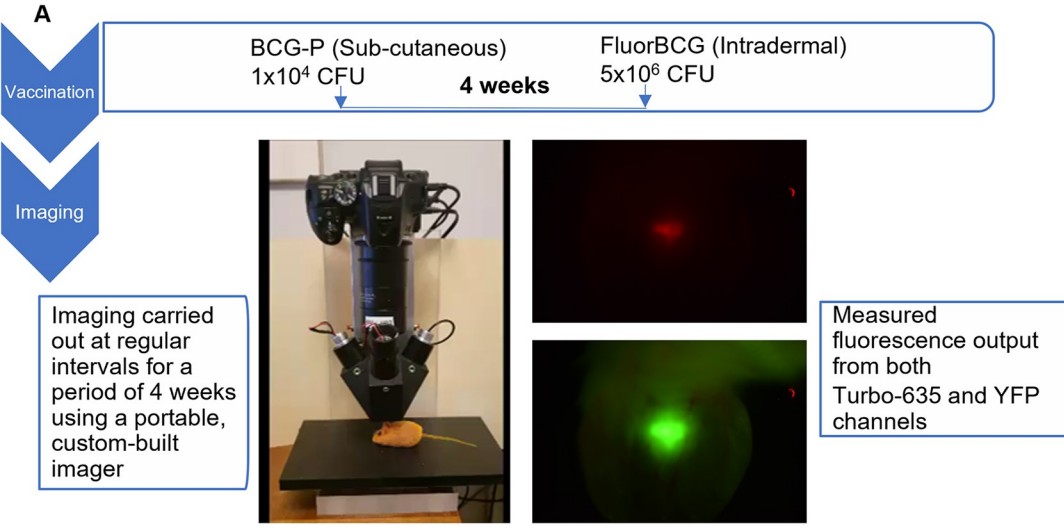

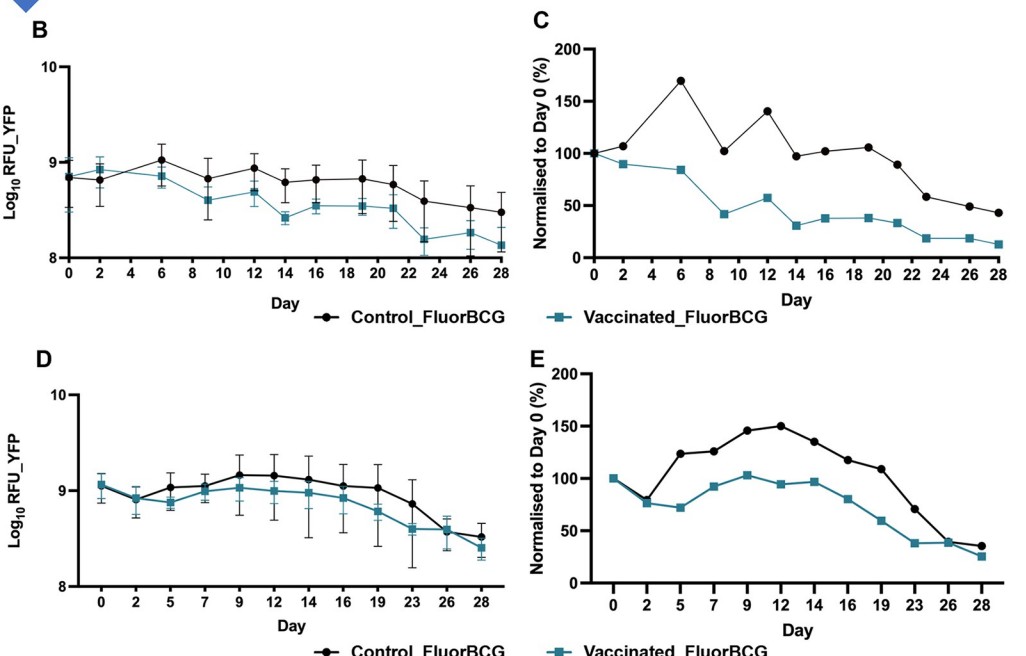

**Fig 1. Testing the skin challenge model using fluorescent BCG.** (**A**) Experimental schema (**B**) and (**D**) raw YFP fluorescence (**C**) and (**E**) normalised YFP fluorescence from both control and BCG vaccinated mice post-ID skin challenge with fluorescent BCG (FluorBCG). Data from 2 experiments are shown. Data represent mean fluorescence ± SD from $n$ = 5 mice (average of 2 ears per mouse). The data underlying this figure can be found in S1 Data. BCG, bacille Calmette-Guérin; ID, intradermal; YFP, yellow fluorescent protein.

replication kinetics with fluorescence output, a time-course experiment was performed measuring the YFP readout (Fig 2A and 2B) and quantifying bacterial load (Fig 2C and 2D) over 21 days. Both vaccinated and control mice had similar fluorescence readout on days 0 and 2; however, BCG-vaccinated mice displayed lower YFP output between days 7 and 21 (Fig 2A). The trend was mirrored in the readout from the Turbo635 channel (S4 Fig). The effect of BCG immunisation is more evident post-normalisation, with the fluorescence readout being lower in the vaccinated mice in comparison to the controls (Fig 2B).

To link bacterial viability with fluorescence output, bacterial numbers were enumerated in the ear at days 2, 7, and 21 post-challenge (Fig 2C). There is a decrease in the CFUs after day 2, with a further decrease in the vaccinated animals between days 7 and 21, which parallels the onset of adaptive immunity.

It is well documented that BCG is trafficked by host cells from the site of injection to the draining lymph nodes [32–37].To confirm movement of FluorBCG from the ear, bacterial load was quantified in the proximal auricular draining lymph nodes at day 21 post-challenge and showed that the BCG-vaccinated mice had lower bacteria load than the control group, although this was not statistically significant (Fig 2D).

## Antigen presenting cells dominate the local immune environment in the mouse ear following BCG vaccination

To study the cellular milieu in mouse ears after FluorBCG challenge, a panel of defined markers was used to characterise the different cellular subsets by flow cytometry. To obtain single-cell suspensions, mouse ears were enzymatically digested and passed through a cell strainer to obtain viable cells. The gating strategy used for flow cytometry was adapted from Yu and colleagues' study (S5 Fig) [38]. The ear skin consists of several types of innate immune cells including neutrophils, macrophages, dendritic cells, T cells, and mast cells. The local environment of the mouse ear at day 7 following FluorBCG challenge of control animals is dominated by neutrophils, Langerhans cells, and macrophages (Fig 3A). In response to intradermal injection of FluorBCG in control animals, neutrophils in the ear increased to a maximum of 5.79% on day 7, decreasing to 1.74% by day 14 (Fig 3A and 3B). In vaccinated mice challenged with FluorBCG, the frequency of neutrophils at day 7 (3.13%) is lower in comparison to control animals (5.79%), dropping further by day 14 (Fig 3A and 3B). The marker Langerin (CD207) identifies the main subsets of dendritic cells that reside in the mouse ear. Langerhans cells are CD207$^+$, and, although conventionally thought of as belonging to the dendritic cell cohort, recent evidence has redefined these cells to be more like tissue-resident macrophages that have acquired dendritic cell-like functions. Langerhans cells possess both in situ functions and migratory functions that promote antigen presentation and T cell priming [39]. A high frequency of Langerhans cells is present in both control and vaccinated mice with an increase overall in vaccinated mice between day 7and day 14 (Fig 3A and 3B). Localisation of Langerhans cells in the mouse ear corroborates the known prevalence of this subset in the skin epidermis [39]. Another important subset of cells in the skin are the dermal dendritic cells, which are subdivided into CD207$^-$CD103$^{+/-}$ and CD207$^+$ CD103$^+$. Both these subsets of dendritic cells are up-regulated in response to BCG vaccination, with the vaccinated mice exhibiting a large influx of CD207$^+$ dermal dendritic cells (Fig 3A and 3B) in comparison to the control mice. In response to BCG vaccination, there is an influx of macrophages that persist in the skin until day 14 (Fig 3A and 3B).

These results suggest that the response to FluorBCG intradermal challenge leads to an inflammatory influx dominated by migratory dendritic cells, neutrophils, and macrophages.

Next, we investigated if viable FluorBCG can be recovered from selected subsets of immune cells from the ear. Neutrophils, macrophages, and dendritic cells were gated for the presence

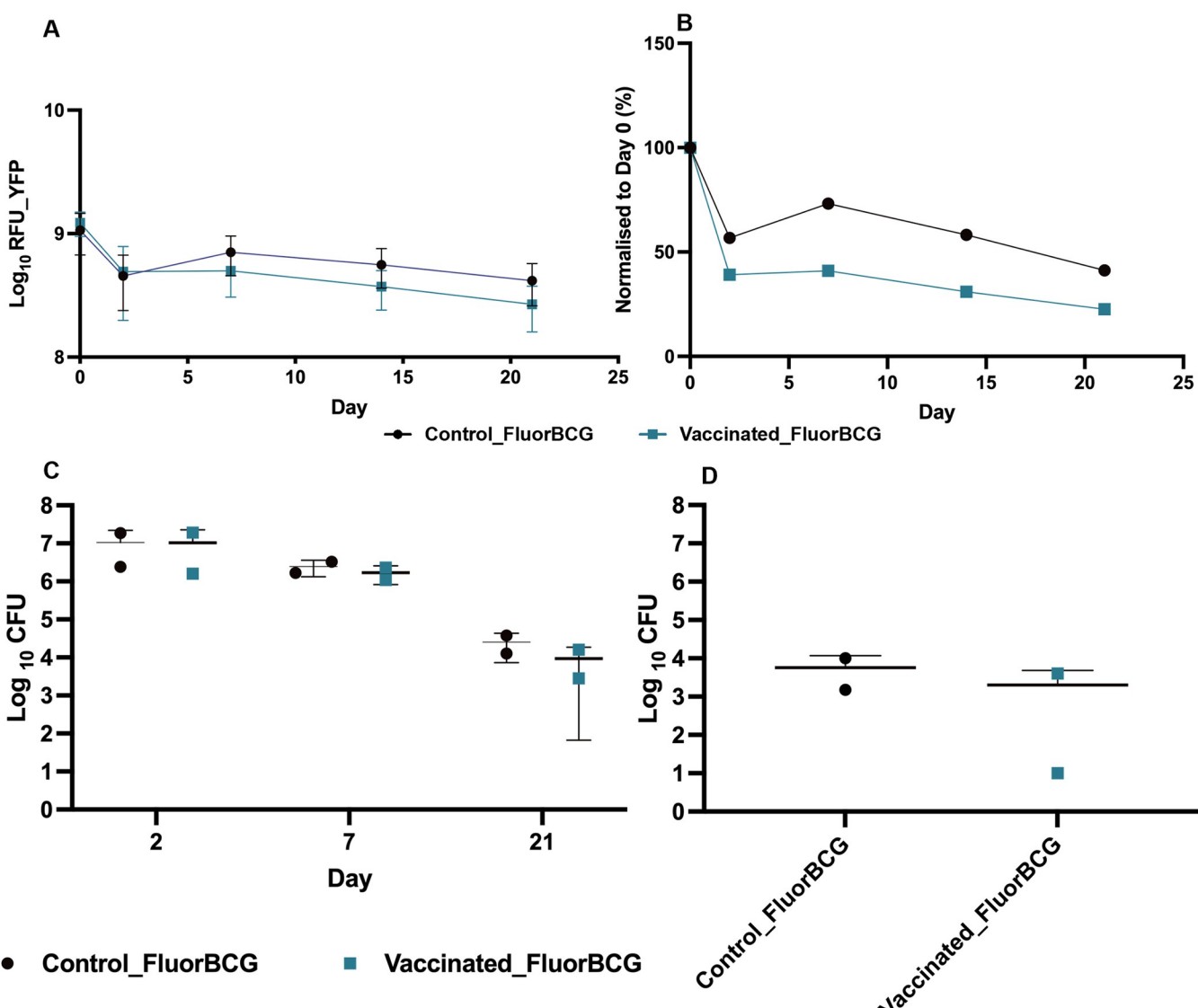

**Fig 2. Quantification of bacterial load in the ear and lymph nodes of mice challenged with fluorescent BCG.** Mice were imaged and fluorescence from (**A**) YFP channels were quantified and normalised to day 0 (**B**) following intradermal skin challenge with fluorescent BCG. Data represent mean fluorescence ± SD from $n$ = 5 mice. The bacterial load was quantified in the ears (**C**) and in the lymph nodes (**D**). The ears and proximal auricular draining lymph nodes from $n$ = 5 mice were pooled, and CFUs were quantified at various time points. Data from one of 2 experiments are shown (**A** and **B**) and average of 2 experiments (**C** and **D**). Error bars represent mean ± SD. The data underlying this figure can be found in S1 Data. BCG, bacille Calmette-Guérin; CFU, colony-forming unit; YFP, yellow fluorescent protein.

of YFP$^+$BCG cells, single-cell sorted using a FACS Aria sorter, and plated to determine CFUs. Neutrophils (CD45$^+$Ly6G$^+$) accounted for a minor population of cells in the ear (Fig 4A and 4B) but had a high BCG burden, which was comparable to the macrophage and dendritic cells present in the ear at day 7 (Fig 4E) with a reduced bacterial load on day 14 (Fig 4F). The reduction in viable bacterial numbers in neutrophils noted at day 14 (Fig 4F) does not match with the expression of YFP$^+$ bacteria in neutrophils (Fig 4D); however, the frequency of YFP+ neutrophils does match with frequency of YFP+ macrophages and dendritic cell subsets. In contrast, macrophages were the dominant population in the mouse ear (Fig 4A and 4B), but only 12% of this subset carried YFP$^+$ BCG (Fig 4C and 4D). Overall, reduction of the viable bacterial load was observed by day 14 in neutrophils, macrophages, and dendritic cells (Fig 4F).

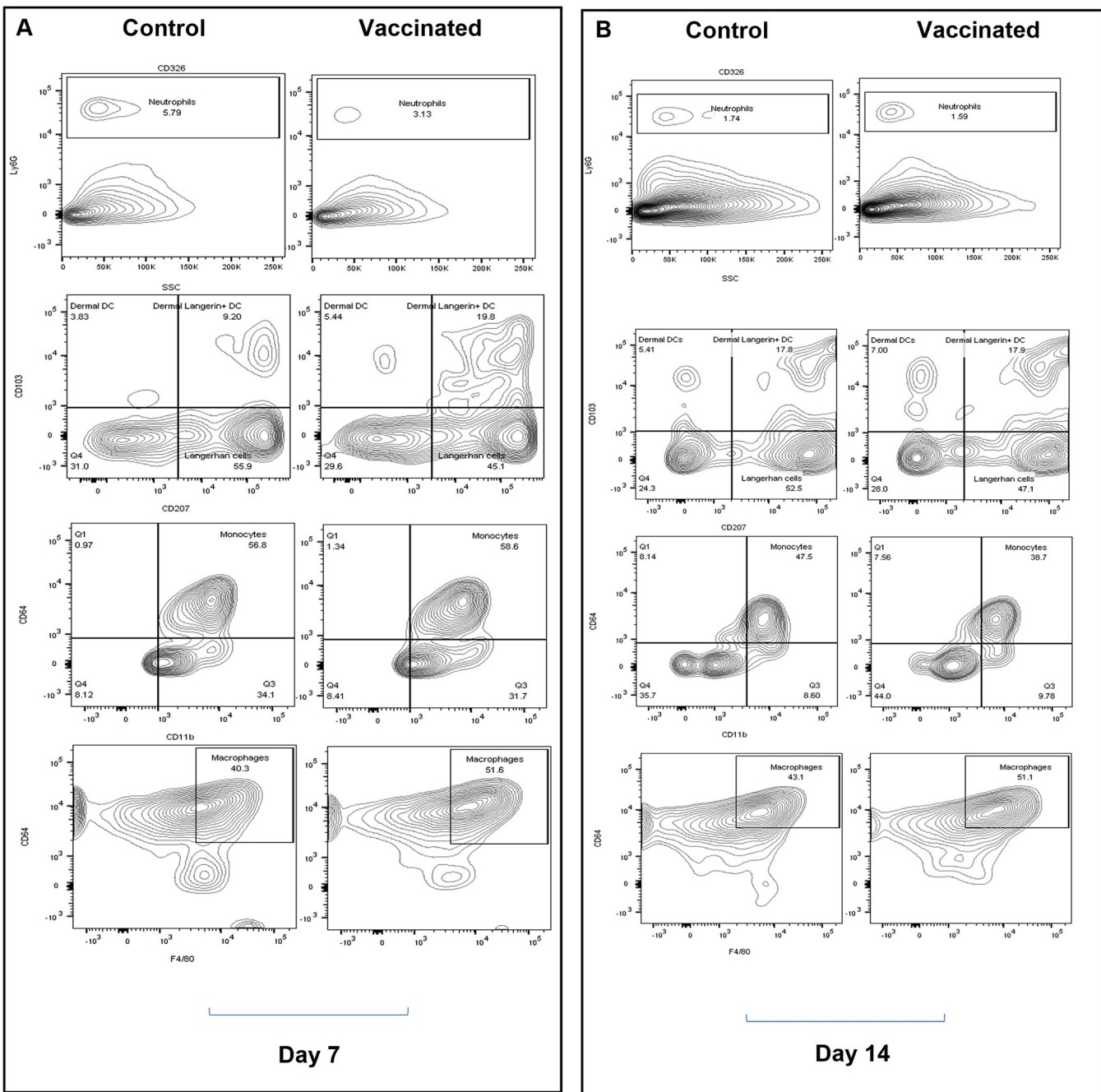

**Fig 3. Immunophenotyping of cellular subsets in the mouse ear.** Mice were immunised with BCG and post-4-week rest, mice were challenged ID in the ear with FluorBCG and cellular infiltrates at the site of challenge were measured at day 7 (**A**) and day 14 (**B**). Gates in contour plots contains single-cell populations—neutrophils (CD45$^+$Ly6G$^+$), dendritic cells subdivided into Langerhans cells* (CD45$^+$MHC-II$^+$CD207$^+$CD103$^-$), dermal DCs (CD45$^+$MHC-II$^+$CD207$^-$CD103$^+$) and dermal langerin$^+$ DCs (CD45$^+$MHC-II$^+$CD207$^+$CD103$^+$); monocytes (CD45$^+$CD11b$^+$CD64$^{int}$) and macrophages (CD45$^+$MHC-II$^+$F4/80$^+$CD64$^+$). *Langerhans cells, although classified under the dendritic cells group here, are more like tissue-resident macrophages, which acquire a phenotype-like dendritic cells. The data underlying this figure can be found at https://doi.org/10.5281/zenodo.12794251. BCG, bacille Calmette-Guérin; DC, dendritic cell; ID, intradermally.

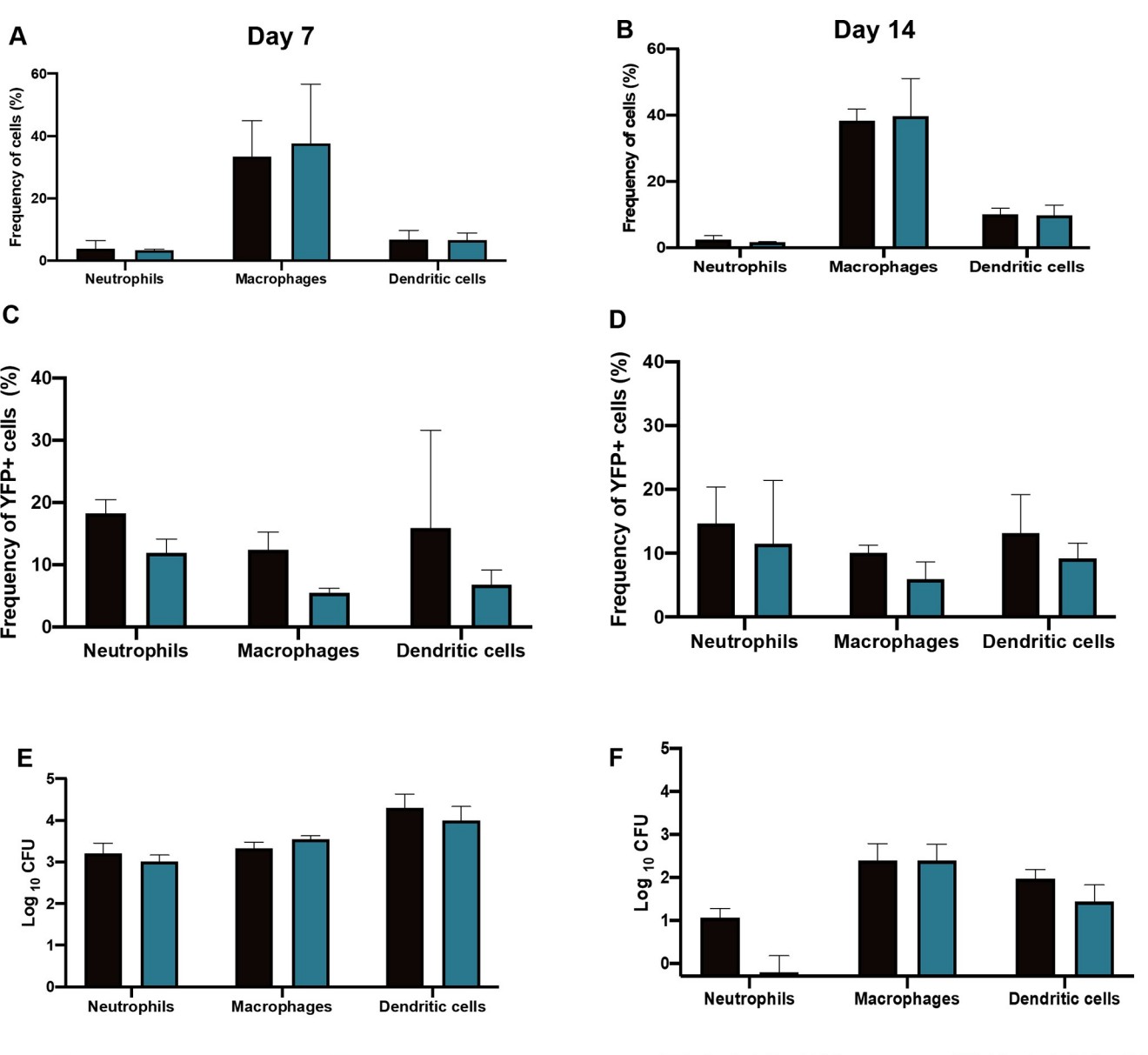

**Fig 4. Enumerating fluorescent BCG associated with the immune subsets in the mouse ear.** Bar graphs (**A**, **B**) depicting immune cell profiles of neutrophils (expressed as a frequency of CD45+ cells), macrophages, and dendritic cells (expressed as a frequency of the myeloid cell population). Immune cellular subsets were further gated on YFP+ cells to isolate populations that phagocytosed fluorescent BCG (**C**, **D**). These gated populations were sorted and plated to quantify viable BCG (**E**, **F**). Flow cytometry data are an average of 2 independent experiments with *n* = 5 mice per group. Error bars represent mean ± SD. The data underlying this figure can be found in S1 Data. BCG, bacille Calmette-Guérin; CFU, colony-forming unit; YFP, yellow fluorescent protein.

## Fluorescence readout from the skin relates to protective immunity in the lung

The ideal route of challenge in humans would be the pulmonary route as it mimics the natural route of infection with Mtb but detecting challenge bacteria in vivo in the lung poses obstacles. Exposure via the skin is a more feasible approach, and detection of the bacteria can be achieved by noninvasive methods. It has been reported that the efficacy of BCG vaccination against intradermal BCG skin challenge has comparable outcomes to an aerosol Mtb challenge,

highlighting the viability of using a BCG-based skin challenge as an alternative to a pulmonary challenge [34]. To test whether the reduction in skin fluorescence in response to BCG vaccination matched an immune response in the lungs, mice were BCG vaccinated, challenged with FluorBCG in the ear, and given a pulmonary Mtb challenge by the intranasal route. In response to BCG vaccination, there was a reduced YFP fluorescence readout overall from the skin of vaccinated mice (Fig 5A and 5B) and Turbo635 (S6 Fig). In the lungs and spleen, BCG vaccination significantly reduced the bacterial burden in comparison to the control group (Fig 5C and 5D). The group of control mice that received a fluorescent BCG challenge (Unvaccinated_FluorBCG_H37Rv) in the ear had a statistically lower bacterial load in the lungs and spleen in comparison to the control mice (Unvaccinated_H37Rv) that did not receive a fluorescent BCG inoculation (Fig 5C and 5D), suggesting the single-dose BCG challenge in the ear affords a degree of protection. The results show that the reduced fluorescence readout from FluorBCG in the skin provides a sensitive and reproducible measure of a relevant biological effect that is shown to reflect traditional measures of TB vaccine efficacy in the lung.

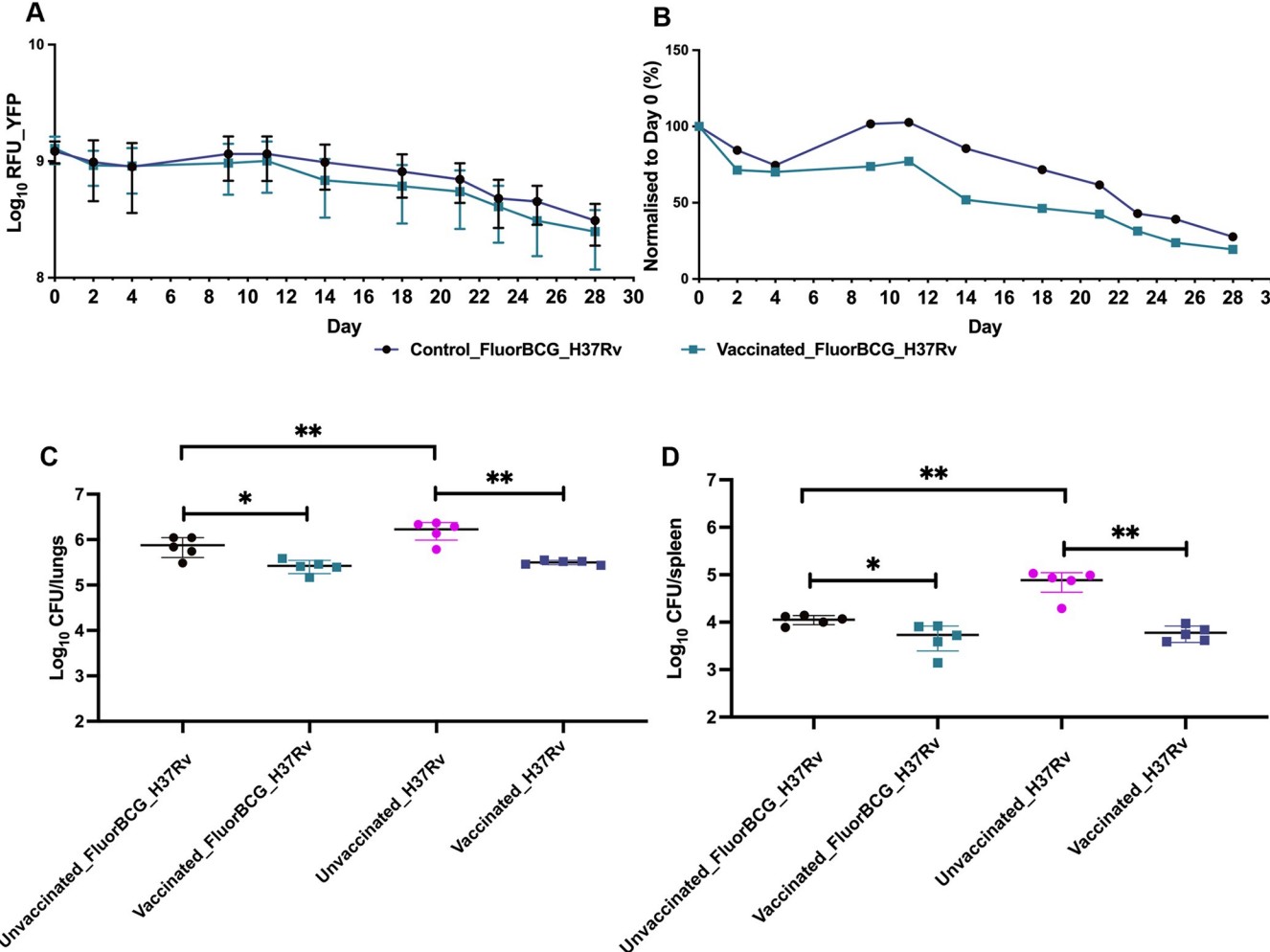

**Fig 5. Fluorescence output from the ear and bacterial burden in the lungs.** BALB/c mice were immunised with a single dose of BCG and after a 4-week rest period, mice were either challenged ID in the ear with $5 \times 10^6$ CFU of FluorBCG and IN with $1 \times 10^3$ H37Rv or only challenged IN with $1 \times 10^3$ H37Rv (H37Rv). Fluorescence intensities from the site of ID challenge in the ear are represented as raw outputs from the YFP channel (**A**) and fluorescence values normalised to day 0, YFP (**B**). Lungs and spleen were harvested 4 weeks post-challenge and processed to quantify the bacterial burden (**C** and **D**). Representative data from duplicate experiments are displayed. Data represent mean ± SD from $n$ = 5 mice. The data underlying this figure can be found in S1 Data. BCG, bacille Calmette-Guérin; CFU, colony-forming unit; ID, intradermally; IN, intranasally; YFP, yellow fluorescent protein.

## Evaluating the skin challenge model using a novel TB subunit vaccine candidate

The skin challenge model was utilised to assess the efficacy of a novel vaccine candidate, ChAdOx1.PPE15. This subunit vaccine candidate comprises a replication-deficient, recombinant chimpanzee adenovirus vector (ChAdOx1), which expresses the mycobacterial antigen PPE15 [24]. Subunit vaccines are primarily designed to boost protective immune responses conferred by the BCG vaccine, but it has been shown that administering ChAdOx1.PPE15 as a single intranasal dose followed by a Mtb challenge resulted in a significant reduction of bacteria load in the lungs compared to controls [16,24]. We immunised BALB/c mice with a single intranasal dose of ChAdOx1.PPE15 and 4 weeks later challenged them with intranasal Mtb and intradermal FluorBCG in the ear. The fluorescence output from immunised mice was stable until day 16 after which there was a decline in the YFP readout. A diminished fluorescence output from the ChAdOx1.PPE15 immunised group in comparison to the control group was noted (Fig 6A). However, the YFP (Fig 6B) and Turbo635 (S7 Fig) normalisation data did not

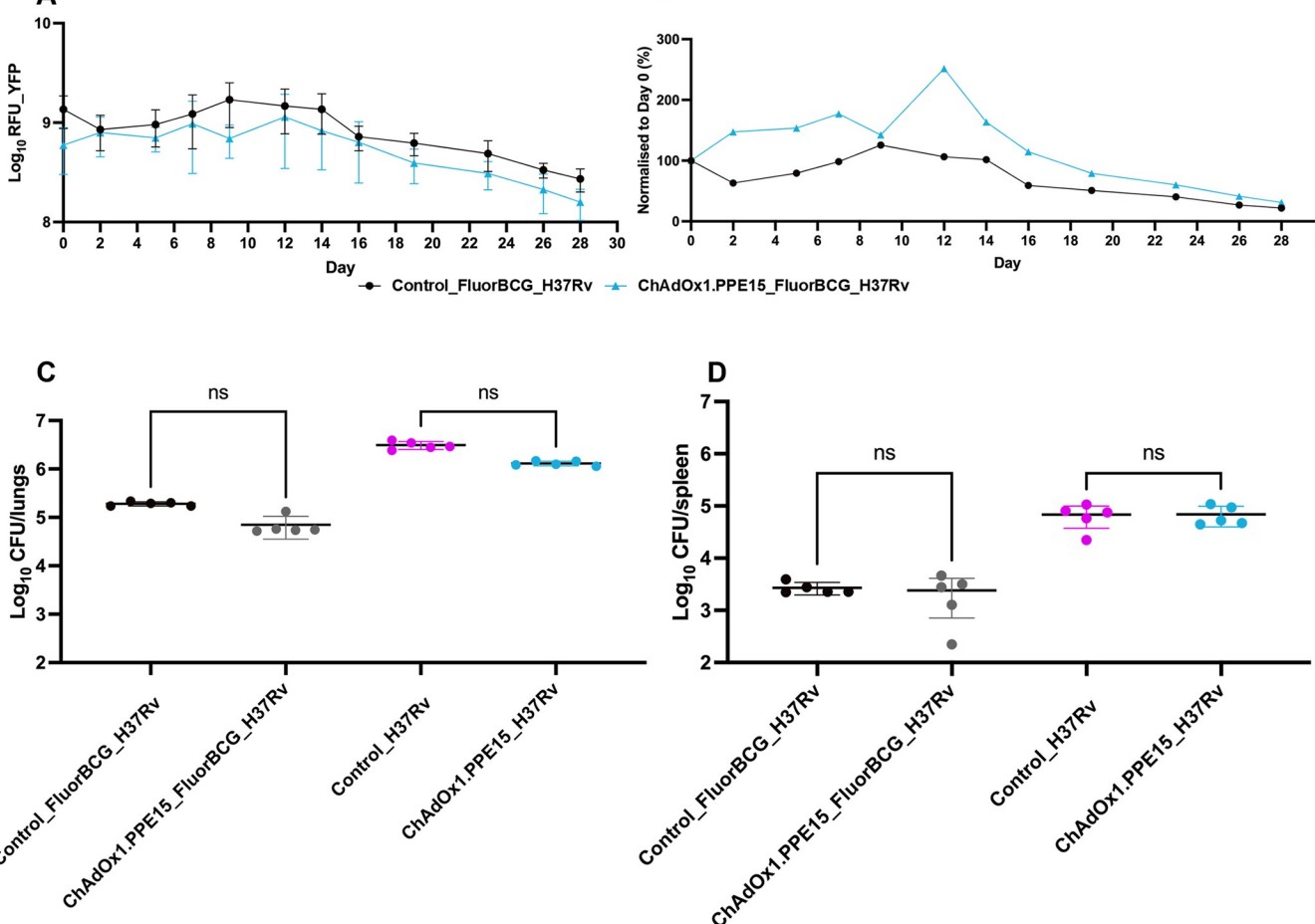

**Fig 6. Evaluating the skin challenge model using a novel vaccine candidate.** BALB/c mice were immunised with a single intranasal dose of ChAdOx1.PPE15, and, after a 4-week rest period, mice were either challenged ID in the ear with $5 \times 10^6$ CFU of FluorBCG and challenged IN with $1 \times 10^3$ H37Rv (H37Rv). Fluorescence intensities from the site of ID challenge in the ear are represented as raw outputs from the YFP channel (**A**) and fluorescence values normalised to day 0, YFP (**B**). Lungs and spleen were harvested 4 weeks post-challenge and processed to quantify the bacterial burden (**C** and **D**). Data represent mean ±SD from $n$ = 5 mice. The data underlying this figure can be found in S1 Data. CFU, colony-forming unit; ID, intradermally; IN, intranasally; YFP, yellow fluorescent protein.

indicate a vaccine effect in mice immunised with only ChAdOx1.PPE15 (Fig 6B). Normalisation of data to day 0 is critical as it removes any confounding variables such as differences in initial dosing. Mice immunised with ChAdOx1.PPE15 also failed to demonstrate a protective effect in the lungs and spleen (Fig 6C and 6D), which corroborates the fluorescence data (Fig 6A and 6B), and mirrors the poor responses reported [24]. A reduction in bacterial load in both lungs and spleen were primarily noted in the groups that received a dose of FluorBCG in the ears (Fig 6C and 6D). Comparison of ChAdOx1.PPE15 vaccinated with BCG vaccinated groups showed no overall difference in terms of RFU (S10 Fig). BCG performed better than ChAdOx1.PPE15 at controlling Mtb H37Rv CFU in spleen but not lungs (S11 Fig, panels A and B); ChAdOx1.PPE15 showed better control of lung CFU but not spleen CFU in the groups that also received intradermal challenge with FluorBCG (S11 Fig, panels C and D).

## Statistical analysis of aggregated experimental data

We built nonlinear mixed-effects models of the aggregated data collected from 5 independent experiments. Based on the graphical and numerical analysis, the baseline fluorescence of YFP in 1 mouse was excluded as it was approximately 100 times lower compared to the median value.

The final model for YFP fluorescence consisted of a biexponential structural model that provided a better fit to the data than a monoexponential model. In addition, a biexponential model was deemed more suitable to describe the initial rapid decline of fluorescence for some groups of mice, which was confirmed by comparing the ability of the mono- and biexponentials models to describe the data through stratified VPCs. IAV on the second intercept and the slope were statistically significant; no covariance between IAV parameters were supported by the data. Box-cox transformation on individual error (eta) distributions were not supported. An additive residual error model was found to be adequate. Covariates in the SCM procedure were implemented as percentage change of typical parameter values. Statistically significant covariates were vaccination (1.52% decrease, $p < 0.001$), baseline fluorescence ($10^{\text{th-quantile}} = 2.2\%$ decrease and $90^{\text{th-quantile}} = 1.36\%$ increase, $p < 0.001$), Mtb pulmonary challenge (1.31% increase, $p < 0.001$), and the reporter (3.43% decrease, $p < 0.001$) on the second intercept, reporter on first intercept (24.2%, increase, $p < 0.001$), and reporter on the second SLOPE (52.4% decrease, $p < 0.001$). However, the covariate effect of vaccination on the second intercept was not statistically significantly different from unvaccinated for the ChAdOx1.PPE15 vaccine, and, as such, the group of mice treated with ChAdOx1.PPE15 was treated as unvaccinated.

The structural base model for Turbo635 fluorescence also consisted of a biexponential model. IAV on the first and second intercept were statistically significant, and no covariance between IAV parameters was found to be significant. Box-cox transformation on ETA distributions were not supported, and an additive residual error model was found to be adequate. Statistically significant covariates were vaccination (1.47% decrease, $p < 0.001$), baseline fluorescence ($10^{\text{th-quantile}} = 2.3\%$ decrease and $90^{\text{th-quantile}} = 3.11\%$ increase, $p < 0.001$), and reporter (4.93% decrease, $p < 0.001$) on the second intercept, reporter (12.3% increase, $p < 0.001$) on the first intercept, reporter (45.2% decrease, $p < 0.001$) and baseline fluorescence ($10^{\text{th-quantile}} = 22.3\%$ decrease and $90^{\text{th-quantile}} = 30.3\%$ increase, $p < 0.001$) on the second slope. Similar to the YFP model, the ChAdOx1.PPE15-vaccinated groups was not statistically significantly different from unvaccinated mice.

The final models were considered suitable to describe the data in all different experimental groups based on stratified VPCs (S8 and S9 Figs). In Figs 7 and 8, typical predictions can be found for all the possible covariate combinations for both the models. The final parameter estimates for the 2 models are provided in Tables 2 and 3.

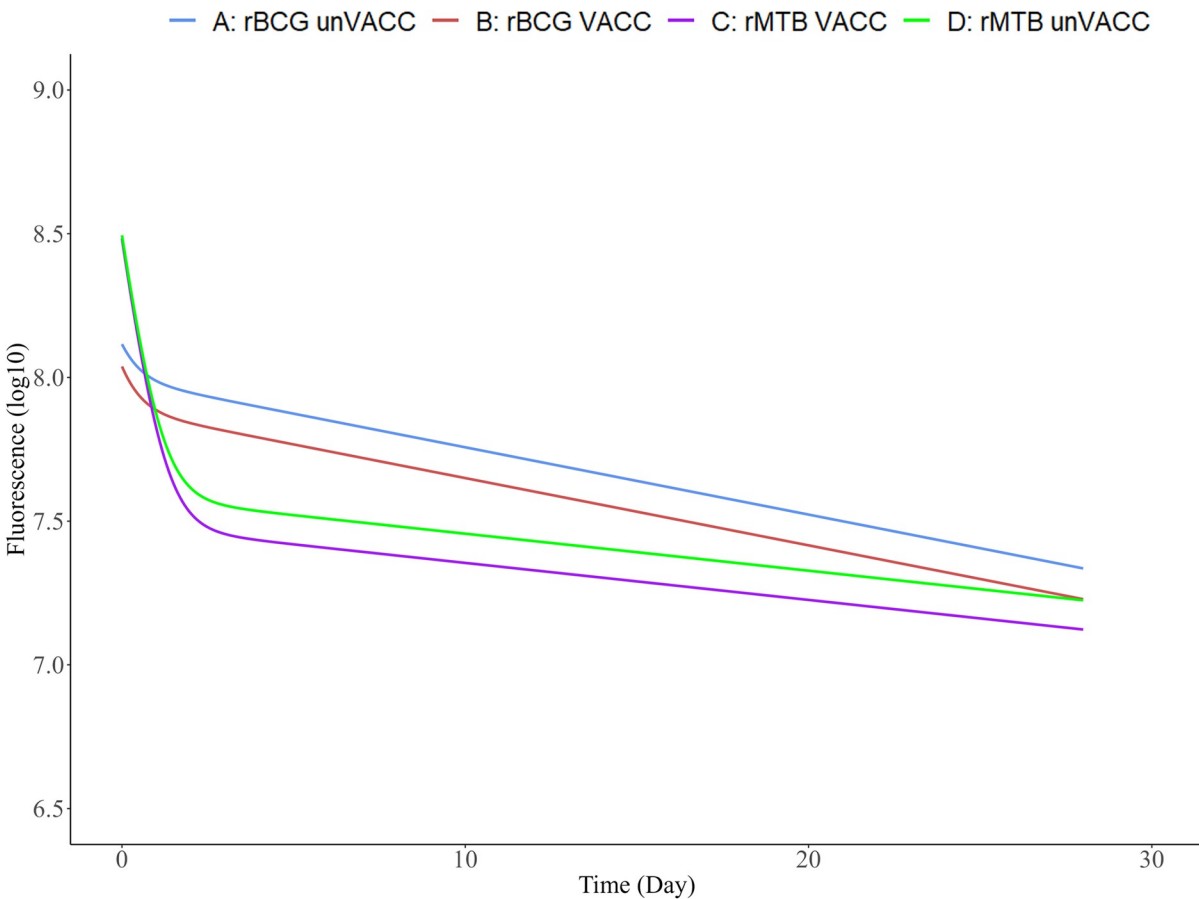

**Fig 7. Visualisation of typical predictions for the identified covariates in the Turbo635 model.** Log10 RFU based on Turbo635 versus time for different combinations of covariate effects. The typical predictions were performed using the final model for Turbo635 (one prediction of the dependent variable per time point, using a fine time grid, over 28 days). This is done for each possible covariate effect. The plots can be recreated by using the parameter values in Table 2 and the covariate relationships described for Turbo635 in the results section.

## Discussion

We report on a proof-of-concept noninvasive, skin-based challenge model for detection and quantification of relevant biological activity reflecting traditional measures of TB vaccine efficacy in the mouse (viable bacterial loads in the lung). Although the utility of the BCG skin challenge model has previously been demonstrated in humans [16], to our knowledge, this is the first reported noninvasive, skin-based challenge study to demonstrate the feasibility of using a fluorescence-based readout as a measure of vaccine efficacy. We also show that the fluorescence output from the skin after challenge with fluorescent BCG serves as a reliable indicator of vaccine-induced immunity in the lung. The fluorescence-based quantitative measurement of protective efficacy described in this study solves several problems associated with vaccine assessment in humans. It uses a proven safe challenge organism, and it allows for repeated noninvasive measurement at the challenge site, providing high-quality temporal data on vaccine responses in vivo. It uses a low-cost imaging platform that can be used in the field, and the technology and approach have potential applications in other areas.

The maximal difference in fluorescence signal between the control and vaccinated groups were observed between day 4 and day 21, providing a window in which new vaccine candidates can be assessed for their efficacy using this noninvasive approach. This defined interval

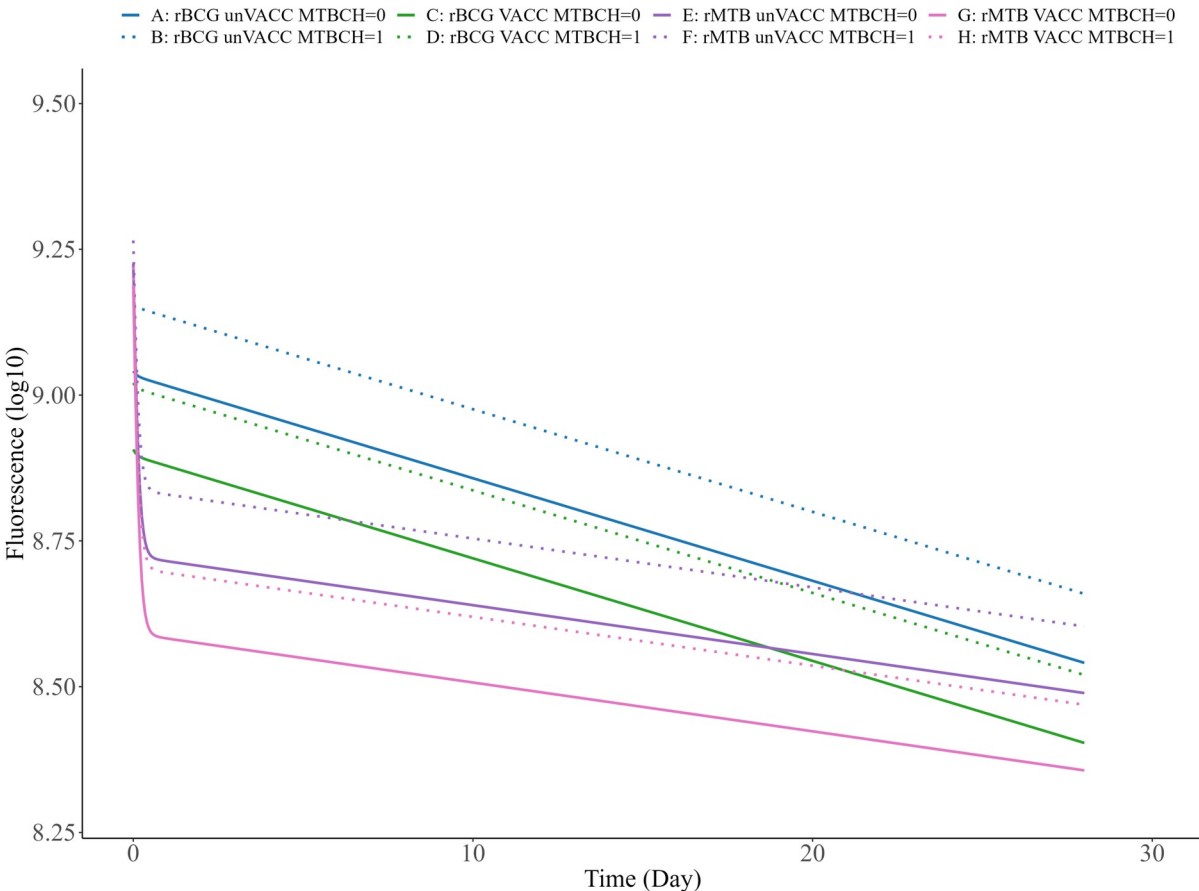

**Fig 8. Visualisation of typical predictions for the identified covariate population in the YFP model.** Log10 RFU based on YFP versus time for different combinations of covariate effects. The typical predictions were performed using the final model for YFP (one prediction of the dependent variable per time point, using a fine time grid, over 28 days). This is done for each possible covariate effect. The plots can be recreated by using the parameter values in Table 3 and the covariate relationships described for YFP in the results section.

for determining vaccine efficacy will cover the period of collective input from both the innate and adaptive immune system, thus providing a more comprehensive output of vaccine protective immunity. This time interval is consistent with the optimum period for detecting live BCG in the skin [34,40]. Although in this first-generation approach, there was some variability in the fluorescence readout, making it harder to achieve statistical significance at specific time points. However, a trend for reduced fluorescence output is noted in the groups of mice vaccinated with BCG compared to control. The relatively long half-life of fluorescent proteins does not seem to impact their utility in this system. The addition of the tripeptide ASV degradation tag to Turbo635 did not change the fluorescent dynamics in this system over time compared to YFP but instead reduced the overall fluorescent signal. Whether Turbo635-ASV is advantageous in other systems remains to be tested and there is potential for fluorescence optimisation.

We used the dorsal surface of the mouse ear as the site for intradermal injection and imaging. Use of the mouse ear as a surrogate for intradermal vaccination site has several advantages: (1) autofluorescence is low, so background fluorescence levels are negligible; (2) ease of accessibility; and (3) repeated imaging of the same area requires no advance preparation, such as hair removal. Using a microfine, 30G insulin needle, we could precisely deliver the dose needed while minimising disruption of the local microenvironment.

**Table 2. Final parameter estimates for the Turbo635 fluorescence decline model.**

| Parameter | Description | Estimate | Relative standard error |
|---|---|---|---|
| Intercept 1 ($\log_{10}$RFU) | First intercept based on $\log_{10}$RFU | 17.3 | 1.2% |
| Slope1 ($\log_{10}$RFU/day) | First slope describing $\log_{10}$RFU decline | 1.99 | 19% |
| Intercept2 ($\log_{10}$RFU) | Second intercept based on $\log_{10}$RFU | 18.2 | 0.3% |
| Slope2 ($\log_{10}$RFU/day) | Second slope describing $\log_{10}$RFU decline | 0.0539 | 3.4% |
| Reporter on Intercept1 | Effect of using Fluor-Mtb reporter on Intercept1 | 0.123 | 12.4% |
| Reporter on Intercept2 | Effect of using Fluor-Mtb reporter on Intercept2 | −0.0493 | 17.3% |
| Baseline fluorescence on Intercept2 | Effect of baseline value of fluorescence (first fluorescence measurement) on Intercept2, continuous covariate | 0.0511 | 17% |
| Vaccination on Intercept2 | Effect of being unvaccinated or vaccinated with ChAdOx1.PPE15. compared to vaccinated with BCG | 0.0149 | 29.4% |
| Reporter on Slope2 | Effect of using Fluor-Mtb reporter on Slope2 | −0.452 | 13.1% |
| Baseline fluorescence on Slope2 | Effect of baseline value of fluorescence (first fluorescence measurement) on Slope2, continuous covariate | 0.574 | 13.1% |
| IAV intercept1 (%CV) | IAV on Intercept1 | 0.00225 | 31.6% |
| IAV intercept2 (%CV) | IAV on Intercept2 | 0.000378 | 13.4% |
| Additive error | Additive residual error | 0.133 | 5% |

BCG, bacille Calmette-Guérin; IAV, inter-animal variability; Mtb, *Mycobacterium tuberculosis*; RFU, relative fluorescence unit.

Live BCG was detectable in the skin till day 21, although BCG counts were not quantified at later time points in this study. In comparison, Minassian and colleagues reported that BCG could be detected until 12 weeks in the mouse ear by culture. However, a significant decline in BCG CFUs was observed after 4 weeks post-immunisation [34]. Furthermore, in a clinical study, live BCG could be cultured from skin punch biopsy specimens up to 4 weeks [16] and 2 months [41] after the BCG challenge. We observed a slight increase in YFP output from

**Table 3. Final parameter estimates for the YFP fluorescence decline model.**

| Parameter | Description | Estimate | Relative standard error |
|---|---|---|---|
| Intercept1 ($\log_{10}$RFU) | First intercept based on $\log_{10}$RFU | 16.8 | 17.4% |
| Slope1 ($\log_{10}$RFU/day) | First slope describing $\log_{10}$RFU decline | 9.99 | 10.1% |
| Intercept2 ($\log_{10}$RFU) | Second intercept based on $\log_{10}$RFU | 20.8 | 0.4% |
| Slope2 ($\log_{10}$RFU/day) | Second slope describing $\log_{10}$RFU decline | 0.0405 | 5.4% |
| Reporter on Intercept1 | Effect of using Fluor-Mtb reporter on Intercept1 | 0.242 | 89.3% |
| Reporter on Intercept2 | Effect of using Fluor-Mtb reporter on Intercept2 | −0.0493 | 17.3% |
| Pulmonary MTB challenge on intercept2 | Effect of being pulmonary infected by Mtb on intercept2 | 0.0131 | 29.4% |
| Baseline fluorescence on Intercept2 | Effect of baseline value of fluorescence (first fluorescence measurement) on Intercept2, continuous covariate | 0.0566 | 16.8% |
| Vaccination on Intercept2 | Effect of being vaccinated with BCG compared to be unvaccinated or vaccinated with ChAdOx1.PPE15 | −0.0152 | 29.4% |
| Reporter on Slope2 | Effect of using Fluor-Mtb reporter on Slope2 | −0.452 | 13.1% |
| IAV intercept2 (%CV) | IAV on Intercept1 | 0.000297 | 19.3% |
| IAV Slope2 (%CV) | IAV on Slope2 | 0.186 | 25.4% |
| Additive error | Additive residual error | 0.0277 | 8.5% |

BCG, bacille Calmette-Guérin; IAV, inter-animal variability; Mtb, *Mycobacterium tuberculosis*; RFU, relative fluorescence unit; YFP, yellow fluorescent protein.

unvaccinated mice between day 2 and day 7; however, this increase in fluorescence readout did not correspond to an increase in CFU counts between day 2 and day 7. Hence, we conclude there was no replication of BCG at the immunisation site between day 2 and day 7. However, a delicate equilibrium may exist between bacterial replication, migration, and bacterial death in the ears, a critical point that needs to be addressed in future experiments. Chambers and colleagues [35] demonstrated that early accumulation of inflammatory cells at the immunisation site reflected the clearance of live BCG from the footpad of mice to the draining lymph nodes. In contrast, dead BCG persisted at the immunisation site for extended periods [35]. We have shown that bacteria migrate out of the ears to the draining lymph nodes with no significant difference in CFU counts between vaccinated and control mice at day 21 post-FluorBCG challenge in the ear. A decline in CFUs was observed by day 7 in both control and BCG-vaccinated mice. By day 21, a reduction in bacterial load was noticed in mice immunised with BCG, which matches the onset of the adaptive immune response and likely to involve more efficient priming of T cells in BCG-vaccinated mice [36].

Two empirical models were successfully developed to describe both fluorophores over time with good predictive performance for both FluorBCG and Fluor-Mtb. For Turbo635, the impact of covariate effects on the fluorescence decay was investigated. The vaccinated population with the FluorBCG reporter exhibits a decline compared to the unvaccinated group (Fig 7A), in addition to reaching a lower fluorescence level. For the Fluor-Mtb reporter, both the unvaccinated and vaccinated groups demonstrate a decline compared to the FluorBCG reporter, with the vaccinated group reaching a lower fluorescence level (Fig 7C) and displaying a decline compared to the unvaccinated group. For the identified covariate effects using YFP data, it was observed that the vaccinated population with the FluorBCG reporter exhibits a decline compared to the unvaccinated group and reaching a lower fluorescence level (Fig 8C). Additionally, the effect of pulmonary infection with Mtb leads to an earlier stabilisation of fluorescence levels and a decline for both unvaccinated and vaccinated (Fig 8D and 8B). In the case of the Fluor-Mtb reporter, both the vaccinated and unvaccinated groups show a decline compared to the FluorBCG reporter, with the vaccinated group (Fig 8G and 8H) reaching an even lower fluorescence level and displaying a decline compared to the unvaccinated group (Fig 8E and 8F). Similar to BCG, the effect of pulmonary infection results leads to an earlier stabilisation of fluorescence levels and a decline for both unvaccinated and vaccinated.

In the typical prediction plots, a rapid decline for the Fluor-Mtb-reporter was also noticed, and from Tables 2 and 3, there is a difference in parameter values between the 2 reporters (FluorBCG and Fluor-Mtb). This could be attributed to the Fluor-Mtb reporter being more antigenically complete, thereby facilitating a quicker initial immunological response. However, the reasons behind the reduction in the second intercept value for the YFP reporter due to pulmonary challenge are uncertain. To the best of our knowledge, there are no physiological explanations for this phenomenon compared to the Turbo635 reporter. It is possible that there are minor outlier values within the experiments involving pulmonary infection or that the inclusion of the Mtb-challenge (MTBCH) factor accounts for variances stemming from factors other than the actual pulmonary infection in the YFP model. The effect of the baseline fluorescence was statistically significant for both the second slope and the second intercept in the case of Turbo635, while only for the second intercept in the case of YFP. This discrepancy may possibly be attributed to a higher variance in the readouts from the Turb635 fluorophore. The empirical models described the data well. However, the evaluations of model performance were performed on the same data that were used for model building. The typical prediction plots (Figs 7 and 8) are specific to the mouse model, which has been shown to be a useful screening tool for early-stage TB vaccines, and, although these specific data have yet to be confirmed in human studies, current evidence is supportive. The models analysed all data over

time from all experiments simultaneously using a mixed effects approach, which is suitable for longitudinal datasets. An alternative would be to select a datapoint of interest and perform group-wise statistical comparisons. However, such an approach is not as informative and does not utilise the full set of generated experimental data.

Our study demonstrates the presence of migratory immune cells, particularly an accumulation of dendritic cells at the site of FluorBCG administration in both control and BCG-vaccinated mice. The skin milieu is complex with multiple migratory immune cell subsets [42,43], so we utilised additional cell surface markers to study the different populations of prominent dendritic cell populations in the ear. In this study, the marked decrease in the fluorescence readout in BCG-vaccinated mice from day 7 likely corresponds to an increase in the presence of migratory immune cells such as dermal langerin+ dendritic cells and, to a lesser extent, the Langerhans cells in the ear. BCG vaccination influences the early appearance of migratory dendritic cells [35]. Migratory immune cells are likely to phagocytose and carry BCG to the draining lymph nodes, although the migration of free bacteria to the lymph nodes cannot be discounted. The dermal langerin+ dendritic cells are known to migrate to lymph nodes in response to infection and inflammation and presents antigens to T cells [44]. These cells are also early responders and are continually replenished from the blood [45]. The presence of a large pool of FluorBCG at the administration site is likely to indicate continuous migration and replenishment of dendritic cells, or persistence in the ear. We observed that the population of dermal langerin+ dendritic cells are maintained in the ear until day 14. Neutrophils were observed in both control and vaccinated mouse skin, with the neutrophil population being smaller in the BCG-vaccinated group. Neutrophils are present in the skin under steady-state conditions, and they are part of the early responders to appear in the skin following sterile injury [46]. Neutrophils can be detected as early as 4 hours following intradermal BCG vaccination [33]. Neutrophils have also been shown to migrate to the draining lymph nodes in response to specific microbial stimuli [47]. Although neutrophils were not the dominant population of cells in the ear, they can phagocytose BCG, as the bacterial load in neutrophils was comparable to the bacterial load enumerated from dendritic cells and macrophages on day 7. Abaide and colleagues also reported on the colocalisation of neutrophils with BCG in the ear and demonstrated that neutrophils could phagocytose BCG present in the skin dermis [33].

By day 14, there was a smaller population of neutrophils present in the ear, with the bacterial load in neutrophils lower than the CFUs in macrophages and dendritic cells. However, the frequency of YFP+ neutrophils present in the skin was similar to YFP+ macrophages and YFP+ dendritic cells. The fluorescence readout is indicative that BCG is present in the neutrophils. However, the lower bacterial load by culture from these cells suggests that at day 14, the neutrophils are predominantly harbouring nonviable BCG. The role of neutrophils in the direct clearance of mycobacteria is not clear [33,48–50]; however, they are known to function cooperatively with dendritic cells and macrophages to orchestrate the killing of mycobacteria [51]. Moorlag and colleagues demonstrated that BCG vaccination could induce trained immunity in neutrophils, which contributed to a more efficient response to *C. albicans* infection by the up-regulation of reactive oxygen species and enhanced expression of degranulation markers [52]. Further work needs to be carried out to confirm if neutrophils play a direct role in mycobacterial killing due to increased neutrophil antimicrobial factors or an indirect role by establishing an early inflammatory response following BCG vaccination in our skin challenge model.

The skin-based challenge model described in this study has a unique advantage for detecting and measuring the protective efficacy by a noninvasive method. Furthermore, repeated sampling of the injection site with minimal discomfort could be carried out, eliminating the need for invasive skin sampling procedures. This study demonstrates that the skin-based

fluorescence output is a measurable indicator of the vaccine protective response in the lung. A reduction in the fluorescence output from BCG-vaccinated group compared to the control mirrors a significant decrease in bacterial load in both the lungs and spleen of BCG-vaccinated mice. Control mice that received FluorBCG ID in the ear also demonstrate a vaccine effect that was expected and agrees with previously published data [36]. Additionally, a BCG boosting effect was observed in response to the intradermal administration of FluorBCG in vaccinated mice, which resulted in further reduction of CFU counts in the lungs and spleen. In our study, we have confirmed and extended on the results demonstrating that protection against a skin-based challenge is associated with a protective vaccine response in the lung [37].

This skin challenge model study is a feasibility study that will ultimately aid the development of a human challenge model for assessment of TB vaccines. Such a model needs to be tested for its utility of measuring responses from vaccines other than BCG. We used a subunit vaccine ChAdOx1.PPE15 to test the skin challenge model with a vaccine other than BCG. In our study, ChAdOx1.PPE15 was not effective in conferring a protective immune response, which was confirmed by both the fluorescence readout from the skin and CFU data from the lungs and spleen of BALB/c mice. This observation agrees with data published by Stylianou and colleagues, demonstrating that in CB6F1 mice, ChAdOx1.PPE15 was only effective in boosting BCG vaccination, and administration of the ChAdOx1.PPE15 vaccine on its own demonstrated a moderate vaccine efficacy in CB6F1 mice but not in BALB/c mice [24]. Significant protective immune responses were observed in the groups that were administered FluorBCG. BCG is a vaccine that induces strong protective immune responses in mice, so any effects of ChAdOx1.PPE15 will likely to be masked unless it can significantly boost the BCG immune response.

## Conclusions

Several animal models exist for TB, but none encompasses the complexity of the spectrum of clinical disease, representing a significant hurdle in translational TB research. For TB, where prognostic animal models are lacking, human challenge trials will be beneficial to identify correlates of immune protection and test vaccine efficacy. TB, a chronic disease caused by a virulent bacterium, does not immediately fit the criteria as a disease for which a human challenge trial can be designed easily. Using alternative approaches to test the efficacy of vaccines is therefore warranted. This skin-based challenge model is one such approach to stratify and select vaccine candidates based on their immune response to accelerate the most relevant candidates along the clinical trial pipeline.

In summary, in this vaccination-challenge model, serially tracking fluorescently labelled bacteria in the skin using a portable imager is a novel noninvasive strategy for measuring vaccine efficacy. This novel detection method has unique advantages for clinical implementation, and this feasibility study establishes the first steps towards developing a safe, tractable, relevant human challenge model for TB.

## Supporting information

**S1 Fig. pCB22-Turbo635-ASV-YFP plasmid map.**
(TIF)

**S2 Fig. Optimisation of BCG dose.** BALB/c mice were infected ID with 3 concentrations of FluorBCG: FluorBCG_High of $5 \times 10^6$ CFU, FluorBCG_Mid of $5 \times 10^5$ CFU, and FluorBCG_Low of $5 \times 10^4$ CFU. (**A**) Raw YFP fluorescence. (**B**) Raw Turbo-635 fluorescence from unvaccinated mice. Data represent mean fluorescence ± SD from $n = 3$ mice (mean of 2 ears

per mouse). The data underlying this figure can be found in S1 Data. BCG, bacille Calmette-Guérin; CFU, colony-forming unit; ID, intradermally; YFP, yellow fluorescent protein.
(TIF)

**S3 Fig. Measuring the Turbo-635 signal from the skin in response to a fluorescent BCG challenge.** (**A**, **C**) Raw Turbo-635 fluorescence. (**B**, **D**) Normalised Turbo-635 fluorescence from both control and vaccinated mice post-ID skin challenge with fluorescent BCG. Representative data from one of 2 experiments are shown. Data represent mean fluorescence ± SD from $n = 5$ mice (average of 2 ears per mouse). The data underlying this figure can be found in S1 Data.
(TIF)

**S4 Fig. Turbo-635 fluorescence output from BCG-vaccinated mice.** Mice were imaged and fluorescence from (**A**) Turbo-635 channels were quantified and normalised to day 0 (**B**) following intradermal skin challenge with fluorescent BCG. Data from one of 2 experiments are shown representing mean fluorescence ± SD from $n = 5$ mice. The data underlying this figure can be found in S1 Data.
(TIF)

**S5 Fig. Gating strategy for flow cytometry.** Total cells were isolated from murine ear, and single cells were identified using the FSC-A and FSC-H plot. The viability of the skin cell suspension was determined using the SSC-A and viability plot. Viable cells were gated to identify the CD45$^+$cells and then neutrophils (CD45$^+$Ly6G$^+$), dendritic cells subdivided into Langerhans cells* (CD45$^+$MHC-II$^+$CD207$^+$CD103$^-$), dermal DCs (CD45$^+$MHC-II$^+$CD207$^-$CD103$^+$) and dermal langerin$^+$ DCs (CD45$^+$MHC-II$^+$CD207$^+$CD103$^+$); monocytes (CD45$^+$CD11b$^+$CD64$^{int}$) and macrophages (CD45$^+$MHC-II$^+$F4/80$^+$CD64$^+$).
(TIF)

**S6 Fig. Fluorescence output from the ear and bacterial burden in the lungs.** BALB/c mice were immunised with a single dose of BCG, and, after a 4-week rest period, mice were challenged ID in the ear with $5 \times 10^6$ CFU of FluorBCG and IN with $1 \times 10^3$ H37Rv (FluorBCG+H37Rv). Fluorescence intensities from the site of ID challenge in the ear are represented as raw outputs from the Turbo-635 channel (**A**) and fluorescence values normalised to day 0 (**B**). Representative data from duplicate experiments are displayed. Data represent mean ± SD from $n = 5$ mice. The data underlying this figure can be found in S1 Data. BCG, bacille Calmette-Guérin; CFU, colony-forming unit; ID, intradermally; IN, intranasally.
(TIF)

**S7 Fig. Testing the skin challenge model with a TB subunit vaccine.** BALB/c mice were immunised with a single dose of ChAdOx1.PPE15, and, after a 4-week rest period, mice were challenged ID in the ear with $5 \times 10^6$ CFU of FluorBCG and IN with $1 \times 10^3$ H37Rv (FluorBCG+H37Rv). Fluorescence intensities from the site of ID challenge in the ear are represented as raw outputs from the Turbo-635 channel (**A**) and fluorescence values normalised to day 0 (**B**). Representative data from duplicate experiments are displayed. Data represent mean ± SD from $n = 5$ mice. The data underlying this figure can be found in S1 Data. BCG, bacille Calmette-Guérin; CFU, colony-forming unit; ID, intradermally; IN, intranasally.
(TIF)

**S8 Fig. VPC based on the final model for TURBO635 data stratified by all existing covariate effects based on 2,000 simulations.** The solid line is the median of the observed data, the shaded area is the 95% confidence interval for the median, and the open blue circles are the observations. The VPC was generated in NONMEM using the PsN VPC command. It can be

regenerated by utilising the software specified in the statistical analysis method section and the S2 Data.
(TIF)

**S9 Fig. VPC based on the final model for YFP data stratified by all existing covariate effects based on 2,000 simulations.** The solid line is the median of the observed data, the shaded area is the 95% confidence interval for the median, and the open blue circles are the observations. The VPC was generated in NONMEM using the PsN VPC command. It can be regenerated by utilising the software specified in the statistical analysis method section and the S3 Data.
(TIF)

**S10 Fig. Measuring the skin fluorescence signal from BCG- and ChAdOx.PPE15-vaccinated mice in response to a fluorescent BCG challenge.** BALB/c mice were either immunised with a single dose of BCG or a single intranasal dose of ChAdOx1.PPE15, and, after a 4-week rest period, mice were challenged ID in the ear with $5 \times 10^6$ CFU of FluorBCG. (**A**, **C**) Raw YFP fluorescence. (**E**, **G**) Raw Turbo-635 fluorescence. (**B**, **D**) Normalised YFP fluorescence from both BCG- and ChAdOx.PPE15-vaccinated mice post-ID skin challenge with fluorescent BCG (**F**, **H**) Normalised Turbo-635 fluorescence from both BCG- and ChAdOx.PPE15-vaccinated mice post-ID skin challenge with fluorescent BCG. Data represent the mean fluorescence ± SD from $n = 5$ mice (an average of 2 ears per mouse). The data underlying this figure can be found in S1 Data. BCG, bacille Calmette-Guérin; CFU, colony-forming unit; ID, intradermally; YFP, yellow fluorescent protein.
(TIF)

**S11 Fig. Comparing the skin challenge model using a novel vaccine candidate and BCG.** BALB/c mice were either immunised with a single dose of BCG or a single intranasal dose of ChAdOx1.PPE15, and, after a 4-week rest period, mice were either challenged ID in the ear with $5 \times 10^6$ CFU of FluorBCG and challenged IN with $1 \times 10^3$ H37Rv (H37Rv) or only challenged with $1 \times 10^3$ H37Rv (H37Rv). Lungs (**A**, **C**) and spleen (**B**, **D**) were harvested 4 weeks post-challenge and processed to quantify the bacterial burden. Data represent mean ± SD from $n = 5$ mice. * $p > 0.05$, ** $p > 0.01$. The data underlying this figure can be found in S1 Data. BCG, bacille Calmette-Guérin; CFU, colony-forming unit; ID, intradermally; IN, intranasally.
(TIF)

**S1 Table. Summary of experiment names and treatments.** Each experiment had 5 mice per group and was measured for both ears; mean ears are tabulated. VC-2, VC-4, and VC-5 had 12 RFU readings taken over 28 days. Flow1 and Flow2 had 5 RFU readings taken over 21 days and were also used for flow cytometry and bacterial load measurements in ears and lymph nodes.
(DOCX)

**S1 Data. Numerical data for plots in all figures.**
(XLSX)

**S2 Data. Details of software and statistical analysis methods to recreate S8 Fig.**
(ZIP)

**S3 Data. Details of software and statistical analysis methods to recreate S9 Fig.**
(ZIP)

## Acknowledgments

Thanks to the TB-Human Challenge Team for their input and discussions. Special thanks to Barry Walker for his constant support for the project.

## Author Contributions

**Conceptualization:** Miles Priestman, Iria Uhía, Thomas M. Baer, Ulrika SH Simonsson, Brian D. Robertson.

**Data curation:** Nitya Krishnan, Miles Priestman, Iria Uhía, Natalie Charitakis, Albin Tranberg, Alan Faraj, Ulrika SH Simonsson.

**Formal analysis:** Nitya Krishnan, Miles Priestman, Iria Uhía, Natalie Charitakis, Thomas M. Baer, Albin Tranberg, Alan Faraj, Ulrika SH Simonsson.

**Funding acquisition:** Brian D. Robertson.

**Investigation:** Nitya Krishnan, Miles Priestman, Iria Uhía, Natalie Charitakis, Izabella T. Glegola-Madejska, Thomas M. Baer, Ulrika SH Simonsson.

**Methodology:** Nitya Krishnan, Miles Priestman, Iria Uhía, Natalie Charitakis, Izabella T. Glegola-Madejska, Thomas M. Baer, Albin Tranberg, Alan Faraj, Ulrika SH Simonsson.

**Project administration:** Brian D. Robertson.

**Resources:** Natalie Charitakis, Thomas M. Baer.

**Software:** Natalie Charitakis, Thomas M. Baer, Albin Tranberg, Alan Faraj, Ulrika SH Simonsson.

**Supervision:** Ulrika SH Simonsson, Brian D. Robertson.

**Writing – original draft:** Nitya Krishnan, Brian D. Robertson.

**Writing – review & editing:** Nitya Krishnan, Iria Uhía, Thomas M. Baer, Albin Tranberg, Alan Faraj, Ulrika SH Simonsson, Brian D. Robertson.

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
