## [Editor Report · Decision Letter 0]

31 Oct 2023

Dear Dr Robertson, 

Thank you for submitting your manuscript entitled "A BCG Skin Challenge Model for Assessing TB Vaccine" for consideration as a Research Article by PLOS Biology. Please note that I am handling your manuscript for my colleague Paula Jauregui whilst she is out of office this week. 

Your manuscript has now been evaluated by the PLOS Biology editorial staff, as well as by an academic editor with relevant expertise, and I am writing to let you know that we would like to send your submission out for external peer review.

*IMPORTANT*

After discussions within the editorial team, we would like to consider your manuscript as a 'Methods and Resources' article at the journal (https://journals.plos.org/plosbiology/s/what-we-publish#loc-methods-and-resources-articles). Upon resubmission (details below), we would be grateful if you could please tick 'Methods and Resources' as the article type in the online submission form. 

Before we can send your manuscript to reviewers, we need you to complete your submission by providing the metadata that is required for full assessment. To this end, please login to Editorial Manager where you will find the paper in the 'Submissions Needing Revisions' folder on your homepage. Please click 'Revise Submission' from the Action Links and complete all additional questions in the submission questionnaire.

Once your full submission is complete, your paper will undergo a series of checks in preparation for peer review. After your manuscript has passed the checks it will be sent out for review. To provide the metadata for your submission, please Login to Editorial Manager (https://www.editorialmanager.com/pbiology) within two working days, i.e. by Nov 02 2023 11:59PM.

Kind regards,

Richard

Richard Hodge, PhD

rhodge@plos.org

On behalf of:

---

## [Decision Letter · Decision Letter 1]

11 Jan 2024

Dear Dr Robertson,

Thank you for your patience while your manuscript "A BCG Skin Challenge Model for Assessing TB Vaccine" was peer-reviewed at PLOS Biology, and please accept my apologies for the time it has taken us to return to you with a decision on your study, partially motivated by the Christmas holiday period. I have taken over the handling of your manuscript during my colleague Richard Hodge's absence form the office, so as to prevent any further loss of time. Your manuscript has now been evaluated by the PLOS Biology editors, an Academic Editor with relevant expertise, and by three independent reviewers, whose reports you will find at the end of this email.

As you will see in the reviewer reports, although the reviewers find the tool and the approach of potential interest, they have also raised a substantial number of crucial concerns. Based on their specific comments and following discussion with the Academic Editor, it is clear that a substantial amount of work would be required to meet the criteria for publication in PLOS Biology. Given our and the reviewer interest in your study, we would be open to inviting a comprehensive revision of the work; however, this would need to thoroughly address all the reviewers' comments in full. A successful revision will need to address the issues raised with the experimental design (such as the need for a longer lag time until challenge to be more relevant, and more meaningful comparisons with other vaccines in human clinical trials), strengthening of the statistical analysis and the reporting throughout, performing experiments a significant number of independent times, providing and analysing area-under-the-curve data, providing data not shown (which we do not allow), limiting conclusions to what is shown, etc.

Given the extent of revision that would be needed, we cannot make a decision about publication until we have seen the revised manuscript and your response to the reviewers' comments. Your revised manuscript would need to be seen by the reviewers again, but please note that we would not engage them unless their main concerns have been addressed. 

We appreciate that these requests represent a great deal of extra work, and we are willing to relax our standard revision time to allow you 6 months to revise your study. Please email us (plosbiology@plos.org) if you have any questions or concerns, or envision needing a (short) extension. Given the nature of the necessary revisions, which represent a substantial amount of work of uncertain outcome, we would also understand if you rather chose to seek more rapid publication elsewhere. Please let us know if this is the case.

**IMPORTANT - SUBMITTING YOUR REVISION**

*Resubmission Checklist*

*Published Peer Review*

*PLOS Data Policy*

*Blot and Gel Data Policy*

Sincerely,

Nonia

Nonia Pariente, PhD

Editor in Chief

PLOS Biology

on behalf of 

Richard

Richard Hodge, 

Senior Editor

PLOS Biology

rhodge@plos.org

REVIEWS:

Reviewer #1: 

The authors describe a present the pre-clinical evaluation of a new tool (fluorescent BCG) for human challenge experiments. The idea and the tool are meritorious, but the presentation of the findings made it a bit challenging to extract the key messages. 

1) Abstract: Starts with human challenge model on line 12. I was expecting human data. Line 25: vaccinated mice. Shouldn't it be stated somewhere in the methods section that this is a mouse study? 

2) Overall experimental plan: There were at least two studies presented, that were partially overlapping. As I read it, they asked two questions: How good is this non-invasive tool as a function of BCG or subunit vaccination? Where are the live bacteria in the mouse ears using FACS? I understand the direct relevance of the first question for human challenges. I do not see the link between the second question and human challenges. If there is an anticipated use case in giving BCG to human volunteers and then cutting out the site of BCG challenge for FACS, this should be stated. This seems to go against the other goal, which is non-invasive monitoring.

3) Statistical analysis section was 2 pages, but in the end, we don't see any data or statistics (p-values or confidence intervals) in the abstract or figures or figure legends. We read that there is a statistically significant effect of vaccination (line 24) but it's not clear whether this is BCG vs. PBS, or also subunit vs. BCG, or BCG vs. subunit. Where was this shown? We read in the abstract that there is a correlation but the figures presented showed that one assay paralleled the results of the other assay (starting line 273). Wouldn't it be possible to calculate the normalized fluorescence over time (compared to day 0) and also calculate the nomalized CFU over time (compared to day 0), then make a scatterplot to show show well these correlated, in the presence or absence of vaccination?

4) Number of experiments, numbers of mice and Figures. Between the methods and results, it was not clear exactly how many mice were infected / challenged in each experiment and how many datapoints one could expect. This might be better done with a Table. For example, on lines 181-183, we read that there were 100 mice, 45 BCG vaccinated, 45 un-vaccinated, 10 ChAd vaccinated. Figure 1 legends says n = 5, representative of two experiments, but there is no statistical analysis stated here. Figure 2 legend says data presented are mean of 5 mice per group, average of 2 ears. Can the authors graphically represent the 5 datapoints for each group in 2a, if possible? I would want to know if the data were Gaussian before using a mean. Assuming a mean is appropriate for 2a and 2b, I would want to know what the statistical analysis was here comparing the curves and whether it was significant, as stated in the abstract. For 2c and 2d, I think what is shown are the pooled ears from exp 1 and the pooled ears from exp 2. The legend says that the graph shows the average of exp 1 and exp 2, I think this should be the averages of each experiment, for clarity. With pooled data such as these, statistical comparisons do not seem likely to show significance. 

5) The authors had two vaccines (BCG and ChAd) and a standard model to evaluate efficacy. They state in the introduction that there a number of vaccines completing pre-clinical assessment and there is a need to prioritize them. They compare BCG to no BCG, and ChAd to control, then note in the discussion that ChAd did not work on its own, as that it is expected to boost pre-existing BCG immunization. This seemed like a missed opportunity to either compare BCG to ChAd, head-to-head, or to compare BCG to BCG + ChAd, to look for better protection. Can this model not be tested for the combination of vaccines that has being studied in human trials, to explore correlations between what is seen with this model and human clinical trial data? 

6) The authors mention data not shown at least 3 times in the manuscript. Please verify with the journal on its policy about such statements. 

Reviewer #2: 

This is an interesting study using a novel, noninvasive fluorescence monitoring technique to evaluate in mice the feasibility of an intradermal BCG challenge model for assessing in vivo TB immunity. They further assessed with this model in vivo TB immunity induced by 2 different vaccines: BCG and ChAdOx1.PPE15. They do demonstrate that previously BCG-vaccinated mice that are partially protected against an aerosolized Mtb challenge also display more rapid clearance of the fluorescent challenge strain compared with control unvaccinated mice. And they develop a pharmacometric model for analysis of the decline in fluorescence and find evidence that statistically significant covariates for the skin challenge strain clearance included vaccination, baseline fluorescence, added Mtb intranasal challenge and reporter. However, there are some potential problems with the model as described and potential overestimation of the model as a biomarker of protective vaccine-induced immunity.

1) BCG vaccination is given 1 month before the skin challenge. Mice vaccinated with BCG have prolonged persistence of replicating vaccine organisms, lasting up to at least 6-12 months. This may explain why only a small increase in the BCG challenge strain was detected between the challenge day and 2 days post-challenge, and then only reductions in BCG challenge strain load were seen between days 7-21. Maximal innate and/or ongoing effector T cell responses induced by ongoing BCG replication may be active at the time of challenge preventing any meaningful increase in the challenge strain load. These responses may be very different from what is needed to measure long-term vaccine-induced protective memory recall responses against a remote challenge with a pathogen that needs to substantially replicate before disease occurs. 

2) In the discussion, on page 25, the authors state that "In our study we have confirmed and extended on the results demonstrating that protection against a skin-based challenge is a predictor of protective vaccine responses in the lung". Although they have seen an association between more rapid skin clearance of the reporter strains and lung protection against Mtb challenge, the current work is far from demonstrating that the skin model alone can be a reliable predictor of mucosal protection. I would remove statements that claim the model is a reliable "predictor" or "correlate" of mucosal protection. At this point I would say the skin model clearance is associated with lung clearance, but whether it can be a direct predictor or correlate are not yet known. 

3) The pharmacometrics model and analyses of covariance are complicated and difficult for this reviewer to adequately evaluate. It would be helpful to have more discussion of what the biostatistical models actually demonstrate and the limitations of the interpretations. 

4) A better schematic of some of the experiments would be helpful. For example, it was difficult for this reviewer to understand without extensive review that some mice were given both a BCG skin challenge and an Mtb IN challenge at the same time which could alter the performance of the skin challenge given alone (in fact, the statistically significant covariance between the skin model responses and added Mtb IN challenge suggest as much). 

5) There are no significant differences shown for direct comparisons of skin load clearance in vaccinated vs unvaccinated responses. For example, no area-under-the curve analyses are reported which would provide a more sensitive endpoint for testing of significant differences between vaccinated and unvaccinated groups. If the model cannot see significant group response differences, it will not be useful for controlled human infection models designed to help select the most promising TB vaccines for advancing beyond phase 1 trials. 

Reviewer #3: 

Comments to authors:

The manuscript entitled "A BCG skin challenge model for assessing TB vaccines" by Krishnan et al. is an innovative study that addresses a serious need for the TB community. Current methods to compare new TB vaccines require the comparisons of BCG-vaccinations to new TB vaccinations followed by natural M. tuberculosis infections, thus requiring large numbers of individuals to enrol in the study. By performing skin challenges with a fluorescent BCG, much smaller numbers of individuals would be needed to compare various vaccine candidates. Thus this work addresses a great need and provides a new tool and approach to assess TB vaccine efficacies. Interestingly, the authors characterize significant differences between the BCG-vaccinated and unvaccinated cellular immune responses that likely correlate with protection. The authors also show a correlation in skin protection with infection in the lung. A number of pointed need to be addressed before publication. 

Specific Comments:

1. Since this is the first report of this portable imager, a photograph of the unit as well as an example of the image that is analyzed, would be helpful to the reader. Such a Figure would be more informative than Figure 1A..

2. The "Fluorophore Selection" paragraph result section is unclear. Although reference 19 demonstrates that the ASV tag makes the Turbo635 unstable, it is unclear why this unstable fluorophore is being used. Were the authors trying to distinguish between bacterial growth inhibition from bacterial cell killing? The authors mention msfYFP but fail to define the monomeric super folded YFP and then do not explain the relevance of the operon they have constructed. Can they distinguish actively growing cells from cells in stationary growth? The choice of the promoters is not explained in this section nor in the materials and methods. Did the authors do experiments with each fluorescent gee separately? That would have been useful to explain why they used both. The observation that the highest dose gave reproducible results when compared to lower doses is not surprising. . 

3. The "Utilising fluorescence output as a measure of vaccine efficacy in a murine skin challenge model" section provides evidence suggesting that the vaccination controls the infection based on the 5 to 10 fold difference in RLU measurements. This is demonstrated in Fig. 1b and repeated in Figure 2a.a. Figure 2c problematic as the authors report a 10 ro possibly 50-fold differences in CFU's between the unvaccinated and BCG-vaccinated controls at day 2. If I am understanding correctly, the author injected 5 X 106 CFUs in each of 10 ears for 5 mice in each group. Since they pooled the samples they should have had % X 107 CFUs in both sets of mice assuming 100% recovery. They may be getting that number for the unvaccinated but it is unclear how many significant digits they have as they do not show the data. Interestingly, there is 10 -fold or more less cells in the vaccinated. This might be suggesting that there is a log of killing demonstrable in the first few days. The data would have been more compelling if they had analyzed the ears from individual mice un the experiment. 

4. Table 1 would be far more useful for the readers if it included what the molecule and immune cells were recognized by the antibody.

5. Did the authors ever analyze the fluorescence in response to TB chemotherapy. This approach could help distinguish killing from growth inhibition. The use of different promoters that express the fluorescence genes could be valuable in future studies. The discussion has several redundancies and should be shortened but this point is worth mentioning. 

6. Figure 2A uses measurements that vary from 100,000,000 to 1.000.000.000. If these are the only ranges measured, would the data be more convincing if reported linearly? Also, can you recorder 2 or 3 significant digits? As mentioned, above 2C, would be more useful if reported with 2 or 3 significant digits and if the data had been obtained for individual ears and not pools. The 2C experiment would need to be reported from three independent experiments since it seems to be reporting an important biological result.

---

## [Decision Letter · Decision Letter 2]

2 Jul 2024

Dear Brian,

Thank you for your patience while we considered your revised manuscript "A BCG Skin Challenge Model for Assessing TB Vaccine" for publication as a Methods and Resources at PLOS Biology. This revised version of your manuscript has been evaluated by the PLOS Biology editors, the Academic Editor and two of the original reviewers and an additional reviewer.

Based on the reviews, we are likely to accept this manuscript for publication, provided you satisfactorily address the remaining points raised by the reviewers. Please also make sure to address the following data and other policy-related requests.

a) We routinely suggest changes to titles to ensure maximum accessibility for a broad, non-specialist readership, and to ensure they reflect the contents of the paper. In this case, we would suggest a minor edit to the title, as follows. Please ensure you change both the manuscript file and the online submission system, as they need to match for final acceptance:

"A noninvasive BCG skin challenge model for assessing tuberculosis vaccine efficacy"

b) Please note that tuberculosis (the disease) should not be capitalized. This needs to be adjusted in the abstract and throughout the manuscript.

c) Thank you for providing the names of the foundations that supported the project. However, we also require the grant number (or grant titles if not available)

Please supply the numerical values either in the a supplementary file or as a permanent DOI’d deposition for the following figures:

Figure 1BCDE, 2ABCD, 4ABCDEF, 5ABCD, 6ABCD, 7, 8, S2AB, S3ABCD, S4AB, S6AB, S7AB, S8, S9, S10ABCDEFGH, S11ABCD

e) Please cite the location of the data clearly in all relevant main and supplementary Figure legends, e.g. “The data underlying this Figure can be found in S1 Data” or “The data underlying this Figure can be found in https://doi.org/10.5281/zenodo.XXXXX”

f) For figures containing FACS data (Figure 3AB and S5), please provide the FCS files and a picture showing the successive plots and gates that were applied to the FCS files to generate the figure. We ask that you please deposit this data in the FlowRepository (https://flowrepository.org/) and provide the accession number/URL of the deposition in the Data Availability Statement in the online submission form

g) Please ensure that your Data Statement in the submission system accurately describes where your data can be found and is in final format, as it will be published as written there.

h) Please ensure that the code generated for the python pipeline is sufficiently well documented and reusable, and that your Data Statement in the Editorial Manager submission system accurately describes where your code can be found.

Please note that we cannot accept sole deposition of code in GitHub, as this could be changed after publication. However, you can archive the code in Zenodo. Once you do this, it will generate a DOI number, which you will need to provide in the Data Accessibility Statement.

We expect to receive your revised manuscript within two weeks. 

*Published Peer Review History*

*Press*

Sincerely,

Melissa

Melissa Vazquez Hernandez, Ph.D.

Associate Editor

PLOS Biology

REVIEWERS' COMMENTS:

Reviewer #1: 

This paper is much improved. Thank you for your attentiveness to the suggestions of the reviewers. 

Some minor points: 

1) Page 33, Figure 1 legend mentions panels a to d. But there are now 6 panels. Please update legend. 

2) Figures 2, 5, 6 show comparable data, especially the CFU plots at the bottom of each. It is a bit difficult to compare across plots with different Y-axes. Would it be possible to present all these plots with the same minimum and maximum, to give the reader a better appreciation of where they are similar and where they differ?

Reviewer #3: 

The authors have addressed all my concerns adequately. I also think they have done so for the other reviewers. The use of the FluoroBCG and the camera unit make provide important new advances for TB vaccine efficacy studies and I thinks this work merits publication in PLoS Biology

Reviewer #4: 

The authors describe a novel approach to assessing TB vaccine efficacy by describing (i) murine intradermal inoculation with (ii) a recombinant BCG strain with a dual fluorescent reporter and (iii) monitoring fluorescence with a portable imager after vaccination. Overall I found this manuscript to describe a valuable contribution as a Method/Resource. 

In this revised version (I did not review the original submission), the authors have attempted to address each of the three reviewers' comments. 

However, I find that there is still a lack of clarity in places. 

Comments

1. In the Abstract, Introduction and indeed Conclusion I found the focus on human infection models surprising given that what is presented in the paper is in fact an elegant reporter system tested in a mouse model. Reviewer 1 also mentioned this, yet the only change was to add a new line in the Abstract: "Here we were report here (sic) on the characterisation of this system in the mouse model". Indeed, the authors criticise mouse models in the Introduction, "TB vaccine discovery has been hampered by animal models that poorly reflect what happens in humans". I would urge the authors to at least mention positive benefits of development and testing in mouse models, given that is what they present. Indeed, a substantial amount of the work presented is characterisation of murine immune cell populations in the ear. Dos this "reflect what happens in humans"?

2. I found the recombinant Fluor-BCG, with the dual reporters and panCD system for plasmid stability, to be very elegant. I would add specific mention of 'Fluor-BCG' to the abstract. 

3. Lines 497-516: Could the authors improve the language in this section, and avoid the multiple uses of vague, non-quantitative phrases such as: "a slightly more initial rapid decline", "a more rapid initial decline", "a slightly more rapid", "somewhat milder initial decline", "a slightly steeper decline", etc. 

4. Line 262: "Quantification showed a 10-fold decrease Turbo635ASV RFUs compared to YFP (Figure S2)". Should this not say "10-fold less Turbo635ASV RFUs" compared to YFP? A 'decrease' suggests it started off high and then reduced compared to YFP; that is not what is shown in Fig S2. 

5. Line 273-74: "The high dose inoculum provided the optimum signal to noise ratio and was well tolerated in mice with no adverse effects (Supplementary Fig S2)". Based on the data presented in the manuscript, it appears only the YFP "provided the optimum signal to noise ratio", and I suggest this is clarified. 

6. Line 251: Add mention of Fluor-BCG "we decided to construct a dual reporter strain, Fluor-BCG,…"

7. Line 307: "To confirm movement of BCG from the ear…". This should be "To confirm movement of Fluor-BCG from the ear…"

8. Line 356: "Fluorescence readout from the skin acts as a relates to protective immunity in the lung". This needs to be reworded. 

9. Figure S4: There is an error in the legend, where "Vaccinated_FluorBCG" is represented by circles in the legend, but graphed as squares in S4(b).

---

## [Editor Report · Decision Letter 3]

25 Jul 2024

Dear Brian,

Thank you for the submission of your revised Methods and Resources "A non-invasive BCG skin challenge model for assessing tuberculosis vaccine efficacy" for publication in PLOS Biology. On behalf of my colleagues and the Academic Editor, Matthew Waldor, I am pleased to say that we can in principle accept your manuscript for publication, provided you address any remaining formatting and reporting issues. These will be detailed in an email you should receive within 2-3 business days from our colleagues in the journal operations team; no action is required from you until then. Please note that we will not be able to formally accept your manuscript and schedule it for publication until you have completed any requested changes.

PRESS

Sincerely, 

Melissa

Melissa Vazquez Hernandez, Ph.D., Ph.D.

Associate Editor

PLOS Biology
